# Ependymal polarity defects coupled with disorganized ciliary beating drive abnormal cerebrospinal fluid flow and spine curvature in zebrafish

Haibo Xie[1,2,3,4]☯, Yunsi Kang[1,2,4]☯, Junjun Liu[1,4], Min Huang[5], Zhicheng Dai[6], Jiale Shi[1,4], Shuo Wang[1,4], Lanqin Li[1,4], Yuan Li[1,4], Pengfei Zheng[1,4], Yi Sun[1,4], Qize Han[1,4], Jingjing Zhang[3,7], Zezhang Zhu[6], Leilei Xu[6], Pamela C. Yelick[8]*, Muqing Cao[5]*, Chengtian Zhao[1,2,4]*

1 Institute of Evolution & Marine Biodiversity, Ocean University of China, Qingdao, China, 2 Laboratory for Marine Biology and Biotechnology, Qingdao National Laboratory for Marine Science and Technology, Qingdao, China, 3 Affiliated Hospital of Guangdong Medical University & Key Laboratory of Zebrafish Model for Development and Disease of Guangdong Medical University, Zhanjiang, China, 4 Fang Zongxi Center, Key Laboratory of Marine Genetics and Breeding, College of Marine Life Sciences, Ocean University of China, Qingdao, China, 5 Key Laboratory of Cell Differentiation and Apoptosis of Chinese Ministry of Education, Department of Pathophysiology, Shanghai Jiao Tong University School of Medicine, Shanghai, China, 6 Division of Spine Surgery, Department of Orthopedic Surgery, Nanjing Drum Tower Hospital, The Affiliated Hospital of Nanjing University Medical School, Nanjing China, 7 The Marine Biomedical Research Institute of Guangdong Zhanjiang, Zhanjiang, China, 8 Department of Orthodontics, Tufts University School of Dental Medicine, Boston, Massachusetts, United States of America

☯ These authors contributed equally to this work.
* pamela.yelick@tufts.edu (PCY); muqingcao@sjtu.edu.cn (MC); chengtian_zhao@ouc.edu.cn (CZ)

**Data Availability Statement:** All relevant data are within the paper and its Supporting Information files.

## Abstract

Idiopathic scoliosis (IS) is the most common spinal deformity diagnosed in childhood or early adolescence, while the underlying pathogenesis of this serious condition remains largely unknown. Here, we report zebrafish *ccdc57* mutants exhibiting scoliosis during late development, similar to that observed in human adolescent idiopathic scoliosis (AIS). Zebrafish *ccdc57* mutants developed hydrocephalus due to cerebrospinal fluid (CSF) flow defects caused by uncoordinated cilia beating in ependymal cells. Mechanistically, Ccdc57 localizes to ciliary basal bodies and controls the planar polarity of ependymal cells through regulating the organization of microtubule networks and proper positioning of basal bodies. Interestingly, ependymal cell polarity defects were first observed in *ccdc57* mutants at approximately 17 days postfertilization, the same time when scoliosis became apparent and prior to multiciliated ependymal cell maturation. We further showed that mutant spinal cord exhibited altered expression pattern of the Urotensin neuropeptides, in consistent with the curvature of the spine. Strikingly, human IS patients also displayed abnormal Urotensin signaling in paraspinal muscles. Altogether, our data suggest that ependymal polarity defects are one of the earliest sign of scoliosis in zebrafish and disclose the essential and conserved roles of Urotensin signaling during scoliosis progression.

**Funding:** This work was supported by the National Natural Science Foundation of China (Nos. 31991194. 32125015 to C.Z., Nos.91954123. 31972887 to M.C., No. 81972029 to Z.Z., No. 32100661 to H.X. and No.32200415 to Y.K.), the Fundamental Research Funds for Central Universities of China (No. 202113046 to Y.K. and No. 201961016 to Y.L.), the the Natural Science Foundation of Shandong Province of China (No. ZR202111120125) to Y.K., Shanghai Science and Technology Commission (20JC1410100) to M.C., and Innovative research team of high-level local universities in Shanghai (SHSMU-ZDCX20211800) to M.C.. PCY was supported by the NIH/NIDCR R01 DE018043 and NIH/NIAMS R21 AR065761. The funders had no role in study design, data collection and analysis, decision to publish, or preparation of the manuscript.

**Competing interests:** The authors have declared that no competing interests exist.

**Abbreviations:** AIS, adolescent idiopathic scoliosis; BB, basal body; BDM, 2,3-butanedione monoxime; CNS, central nervous system; CP, choroid plexus; CSF, cerebrospinal fluid; CSF-cNs, CSF-contacting neurons; DAPI, 4′,6-diamidino-2-phenylindole; dChP, diencephalic choroid plexus; dpf, days postfertilization; hpi, hours postinjection; IS, idiopathic scoliosis; mpf, months postfertilization; MZ, maternal zygotic; PBS, phosphate buffered saline; PCP, planar cell polarity; PEI, polyethyleneimine; PFA, paraformaldehyde; qRT-PCR, quantitative real-time PCR; rPCP, rotational PCP; RF, Reissner fiber; SD, standard deviation; SEM, scanning electron microscopy; tPCP, translational PCP; WGA, wheat germ agglutinin; WISH, whole-mount in situ hybridization.

## Introduction

Idiopathic scoliosis (IS), characterized by the abnormal rotation and curvature of the spine, is the most common spinal deformity, affecting more than 3% of children and adolescents worldwide [1]. Although more than 80% of scoliosis cases are deemed idiopathic, it is believed that genetic factors make significant contributions to the progression of the disease, based on the high incidence of scoliosis in families and twins. Currently, the pathogenesis of IS remains largely unknown due to insufficient knowledge of its etiology and subsequent disease progression.

Recently, the zebrafish has emerged as a powerful model for human scoliosis based on similar spinal column architecture and vertebral structures in zebrafish and humans [2–4]. Moreover, teleost fish exhibit a natural susceptibility to develop spinal curvatures over time, making zebrafish a reliable model for human IS [2,4,5]. Indeed, recent work on several zebrafish scoliosis mutants has provided significant insight into molecular mechanisms regulating scoliosis, including linking scoliosis to cerebrospinal fluid (CSF) flow defects [6]. CSF is a clear fluid that bathes the brain and spinal cord, is crucial for maintaining homeostasis of the central nervous system (CNS), and is produced by specialized ependymal cells in the choroid plexus (CP) of the ventricles of the brain. CSF flow is propelled by the beating of motile cilia, specialized, tiny organelles protruding from the surface of ependymal cells lining the brain ventricles and spinal canal. Zebrafish mutants exhibiting ciliary motility defects often develop hydrocephalus and progressive late-onset scoliosis [6].

In addition to scoliosis, zebrafish ciliary mutants constantly develop body curvature at larval stage [7]. Studies from our lab and other groups have identified Urotensin as the major signaling pathway that functions downstream of motile cilia to regulate body axis development. Mechanistically, cilia-driven CSF flow transmits adrenergic signals to CSF-contacting neurons (CSF-cNs), promoting the synthesis and secretion of urotensin neuropeptides, Urp1 and Urp2. These neuropeptides further activate their receptor, Uts2r3 (previously named Uts2ra), a Urotensin-2 receptor specifically expressed in dorsal slow-twitch muscle fibers. Thus, signals from CSF finally direct dorsal muscle fiber contraction and control proper body axis straightening during early development [8–11].

CSF-cNs are a specialized type of neuron that can sense CSF environmental changes, including pH and osmolarity [12,13]. CSF-cNs also respond to the mechanical signals related to CSF flow or tail bending, thus controling the locomotion of zebrafish larvae [14–17]. CSF-cNs contain highly polarized apical protrusions, which help modulate the mechanical sensory functions of these neurons during spinal curvature [18]. The lumen of the central canal contains a long extracellular thread, the Reissner fiber (RF), which extends from the brain ventricle to the end of the spinal canal. The RF is dynamically formed by the aggregation of SCO-spondin glycoprotein secreted from both the subcommissural organ and the floor plate in zebrafish [19]. CSF flow is essential for RF assembly and zebrafish mutants with ciliary defects are constantly associated with RF loss [20]. Recently, extensive studies have demonstrated critical roles for the RF in mediating CSF signaling and controlling body axis development [11,19–22]. The RF is located in close vicinity to the apical protrusions of the CSF-cNs, which help transduce the mechanical signals from spinal curvature or CSF flow [21]. Zebrafish *scospondin* mutants fail to develop the RF and display severe body curvature defects [19,20,22].

Adrenergic signals are critical for the activation of Urotensin neuropeptides in the CSF-cNs. The RF may provide a scaffold or microenvironment to promote the transduction of CSF adrenergic signals to the CSF-cNs [11,23]. Interestingly, the hypomorphic *scospondin* zebrafish mutants can survive to adulthood and display scoliosis, suggesting a critical role for the RF during later body axis development [10,19,22]. Similarly, zebrafish *uts2r3* mutants also display

severe scoliosis during late development [8]. These works suggest that defects in the Urotensin signaling pathway contribute to scoliosis formation. The Urotensin signaling pathway appears to be conserved in other vertebrates [24], and mutations in *UTS2R*, the human homolog of zebrafish *uts2r3*, is also associated with human scoliosis [25].

CSF is produced by the CP, a highly specialized epithelium located in the ventricles of the brain that are in close contact with ependymal cells [26]. A key feature of brain ventricle ependymal cells is the presence of multiple motile cilia in their apical surface, which need to beat in the same direction to properly propel CSF flow [27]. Planar cell polarity (PCP) signals are essential to define the distribution of these cilia and ensure the proper direction of ciliary beating [28–30]. Of note, ependymal cells display two types of planar polarity—rotational PCP (rPCP) and translational PCP (tPCP)—based on the orientation and positioning of basal body clusters located within cells and tissues [31,32]. Motile cilia are essential for regulating rPCP, while tPCP is established by primary cilia of radial glial cells during differentiation [28,31]. Defects in ependymal cells, including polarity defects, are often associated with hydrocephalus caused by abnormal CSF circulation [29,33].

Congenital hydrocephalus is a common phenotype that occurs in several human disorders including PCD [34]. Intriguingly, PCD patients also exhibit a high prevalence of scoliosis [35], although it remains unclear how hydrocephalus may result in the development of scoliosis. Similarly, hydrocephalus and scoliosis occur in many zebrafish ciliary mutants. Interestingly, spinal curvature did not develop in these ciliary mutants until approximately 3 weeks postfertilization, a similar stage to the beginning of scoliosis in adolescent idiopathic scoliosis (AIS) patients. The molecular mechanisms of scoliosis development at these stages remains to be elucidated. Here, we have characterized a late-onset zebrafish scoliosis mutant exhibiting loss of function of Ccdc57. Zebrafish *ccdc57* mutants develop severe hydrocephalus due to defects in the coordinated beating of multiple cilia. We provide data showing that Ccdc57 regulates ependymal PCP, whose defects are likely the major cause of scoliosis formation in *ccdc57* mutants. Moreover, we describe the relationship between spine curvature and abnormal Urotensin expression caused by abnormal CSF circulation, thereby providing important mechanistic clues for the formation of scoliosis.

## Results

### Mutation of *ccdc57* results in scoliosis in zebrafish

In an ENU-based screen for zebrafish mutants with mineralized craniofacial and skeletal tissue defects, we identified *tft*$^{168N}$ mutants that displayed severe spinal curvatures in adults (Fig 1A). Both Micro-CT and Alizarin red staining showed that *tft*$^{168N}$ mutants displayed abnormal three-dimensional curvatures and deformities of the spinal vertebrae when viewed from both dorsal and lateral positions (Figs 1B and S1A). Interestingly, the severity of the spinal curvatures was comparable in male and female zebrafish mutants, in that the cobb angles measured from dorsal-ventral and medio-lateral curvature showed similar distribution patterns (S1B–S1E Fig). To identify the gene responsible for this phenotype, we performed Next Generation sequencing and identified a mutant locus at the genomic region encoding the *ccdc57* gene. The *ccdc57* gene contains 19 exons encoding 979 amino acids, and the *tft*$^{168N}$ allele introduced a stop codon in the 11th exon of *ccdc57* resulting in the truncation of the final 407 C-terminal amino acids (Fig 1C). To further validate this gene mutation, we generated *ccdc57* mutants via CRISPR/Cas9 methods and recovered two mutant alleles, one with a 2-bp insertion (+2) and another with a 7-bp deletion (Δ7) in the first exon (Fig 1C). Both of these two *ccdc57* mutant alleles displayed scoliosis (Fig 1D). Moreover, complementation testing between the Δ7 and *tft*$^{168N}$ alleles showed that these two alleles failed to complement each other, confirming that

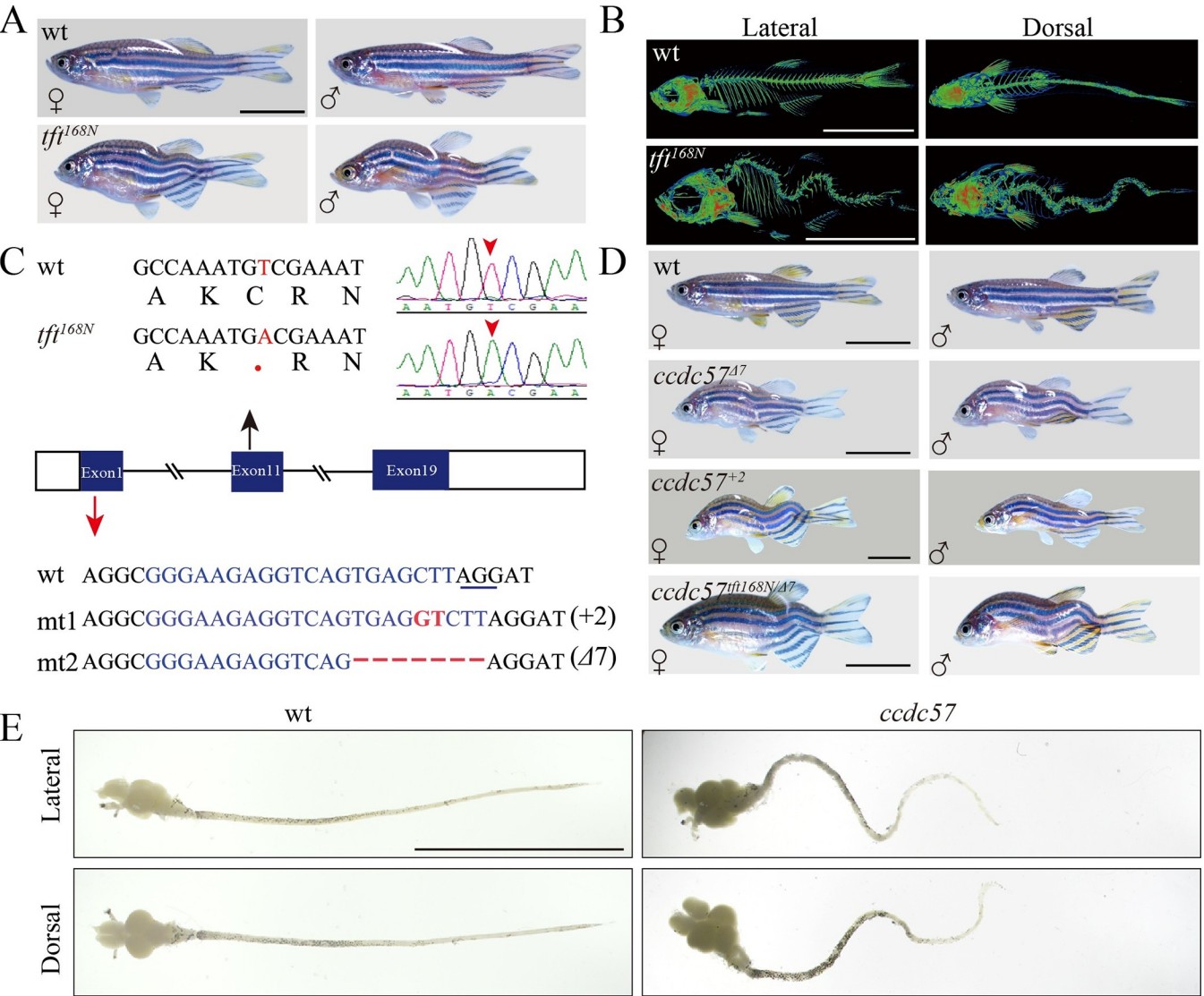

**Fig 1. Mutation of *ccdc57* leads to scoliosis.** (**A**) Representative images of male and female wild type and *tft^168N* mutants. (**B**) Micro-CT images showing lateral and dorsal views of 3-months-old wild type and *tft^168N* mutants. (**C**) Genomic structure and sequences of wild type, *tft^168N* mutant and two *ccdc57* mutant alleles generated with CRISPR/Cas9. The underlined sequence in wt indicates PAM sequence of sgRNA target. (**D**) Representative images of male and female wild type and *ccdc57* mutants with different genetic background as indicated. (**E**) The dissected brain and spinal cord in wild type and *ccdc57* mutant. Scale bars: 1 cm in panels A, B, D, E.

mutations in *ccdc57* were causative for scoliosis in these mutants (Fig 1D). Of note, when dissected from fixed vertebrae, *ccdc57* mutant spinal cords also displayed curvature defects that closely mimicked those of the spinal curvatures (Fig 1E).

To define the progression of scoliosis in *ccdc57* zebrafish mutants, we conducted developmental analyses of the mutant phenotype. Zygotic *ccdc57^Δ7* mutants displayed body curvature at 3 days postfertilization (dpf) (S2A Fig). Interestingly, body curvature was not apparent in 3 dpf *ccdc57^tft168N* mutants (S2A Fig). Such disparity may be due to the presence of truncated forms of Ccdc57 proteins in the *ccdc57^tft168N* mutants, as the *tft^168N* allele encodes the N-terminal 572 amino acids of Ccdc57 (Fig 1C). Moreover, maternal Ccdc57 protein may contribute to early embryonic development. Indeed, maternal zygotic (MZ) *ccdc57^tft168N* mutants

displayed body curvature defects at both 3 and 5 dpf, similar to those of MZ $ccdc57^{\Delta7}$ mutants (S2A Fig). In the following studies, we focused on the $ccdc57^{\Delta7}$ mutant allele to evaluate the function of Ccdc57 in spinal development and scoliosis.

In zebrafish, ciliogenesis defects are a major cause of body curvature development [7]. We therefore examined cilia development in $ccdc57$ mutants. Surprisingly, all of the cilia examined were grossly normal in the $ccdc57$ mutants as visualized using anti-glycylated tubulin antibody (S2B–S2F Fig). Although abnormal notochord differentiation can contribute to the development of congenital scoliosis, we did not detect any notochord defects in $ccdc57$ mutants at early stages, suggesting an independent mechanism of scoliosis progression in the absence of Ccdc57 (S2G Fig). In fact, scoliosis was first detected at approximately 17 dpf in both $ccdc57^{tft168N}$ and $ccdc57^{\Delta7}$ mutants, with initial bending of the mutant spine occurring in the middle portion of the trunk (S2H and S2I Fig).

## Hydrocephalus in *ccdc57* mutants

Recently, abnormal CSF flow has been linked to the development of scoliosis [6,8]. Therefore, to better assess the etiology of scoliosis progression in $ccdc57$ mutants, we dissected whole brains from $ccdc57$ mutants and wild type sibling controls. The $ccdc57$ mutant brain was larger and appeared transparent due to an expanded ependymal epithelium filled with CSF (Fig 2A and 2B). Histological analysis of cross-sectioned brains revealed the presence of dilated ventricles in the optic tectum and rhombencephalon of $ccdc57$ mutants, as well as in the central spinal canal (Fig 2C–2E). In addition, micro-CT analysis confirmed the presence of dilated ventricles in $ccdc57$ mutant brains (Fig 2F).

As described, CSF is produced by the CP. Therefore, we further examined the CP in wild type and $ccdc57$ mutants focusing on the epithelial monolayer connecting the telencephalon and optic tectum known as the forebrain CP or diencephalic CP (dChP) [36–38] (Fig 2G). Three-dimensional reconstructions of the dChP in wild type zebrafish revealed a chapeau-like structure with multiple folds covering the brain tissues (Fig 2H and S1 Movie). In contrast, $ccdc57$ mutant dChP appeared stretched and lacked foldings, due to excess CSF accumulation (Fig 2I and S2 Movie). Next, we investigated whether $ccdc57$ mutant larvae showed signs of hydrocephalus. Injection of Rhodamine- or FITC-conjugated fluorescent beads into 2 or 3 dpf zebrafish larvae revealed no obvious differences in the brain ventricle size between mutants and siblings (S3 Fig). Together, these data suggested that, similar to the late appearance of scoliosis, $ccdc57$ mutants developed hydrocephalus at later stages of development.

## Planar polarity defects of cilia and basal bodies in *ccdc57* mutant ependymal cells

Excess accumulation of CSF in cerebral ventricles is one of the major causes of hydrocephalus. In zebrafish, directed CSF flow is facilitated by the coordinated beating of motile cilia in multiciliated ependymal cells lining the brain ventricles. Focusing on the telencephalon and dChP regions, we identified multiciliated cells largely restricted to the central portion of the ependymal layer (S4 Fig). Next, we monitored motile cilia beating in these ependymal cells. In adult wild type zebrafish, all motile cilia bent to the direction of the fluid flow, and multicilia bundles beat synchronously (Fig 3A and S3 Movie). In contrast, multicilia bent in a variety of directions in $ccdc57$ mutants (Fig 3A and S4 Movie). Even within the same ciliary bundle, individual cilia beating appeared disorganized (S4 Movie). We next performed scanning electron microscopy (SEM) analyses to visualize the ultrastructure of the ependymal cilia bundles. Compared to the clustered distribution in wild type fish, ciliary bundles appeared highly disorganized in $ccdc57$ mutants (Fig 3B). Aberrant distribution of these ependymal cell ciliary bundles was

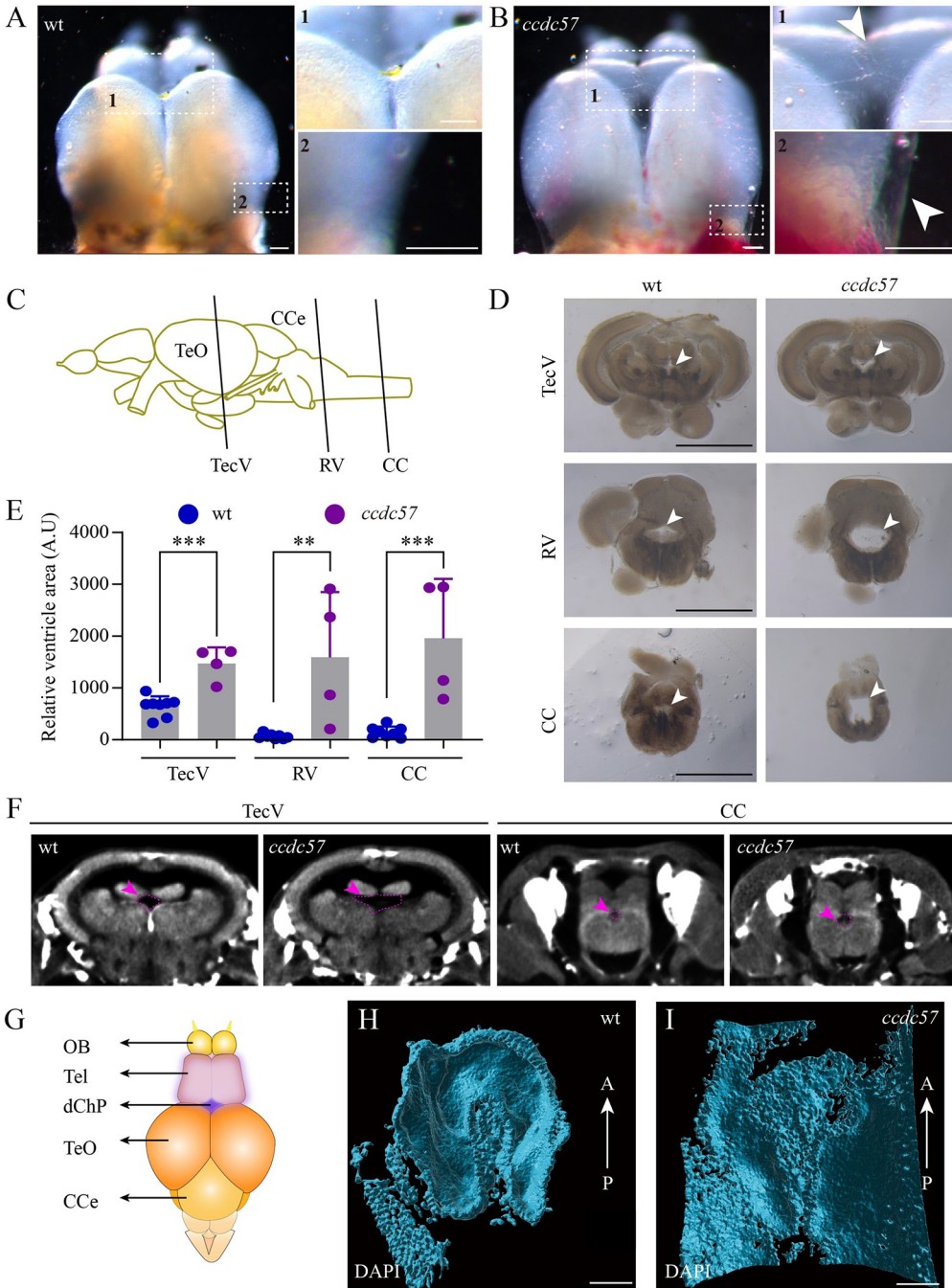

**Fig 2. Hydrocephalus in *ccdc57* mutants.** (**A**, **B**) External phenotypes of telencephalons in wild type and *ccdc57* mutant. Arrows point to the expanded ependymal epithelia due to excess CSF accumulation. (**C**) Diagram of the brain tissue in zebrafish showing the sites of crossing section in different ventricles. CCe, corpus cerebelli; TeO, tectum opticum; CC, central canal; TecV, tectal ventricle; RV, rhombencephalic ventricle. (**D**) Histological cross-sections showing ventricles (arrows) at different sites as indicated. (**E**) Bar graph showing relative size of brain ventricles at different sites as indicated. (**F**) Micro-CT images showing the ventricles (arrows) in TecV and CC of wild type and mutant. (**G**) Diagram showing the position of diencephalic choroid plexus (dChP) in adult zebrafish. OB, olfactory bulb; Tel, telencephalon. (**H**, **I**) Three-dimensional reconstruction of dChP visualized with DAPI staining in wild type and *ccdc57* mutant. These still images are from S1 Movie and visualized from ventral side of the dChP. A, anterior; P, posterior. Scale bars: 100 μm in panels A and B, 1 mm in panel D, and 50 μm in panels H and I. The data underlying the graphs shown in the figure can be found in S1 Data.

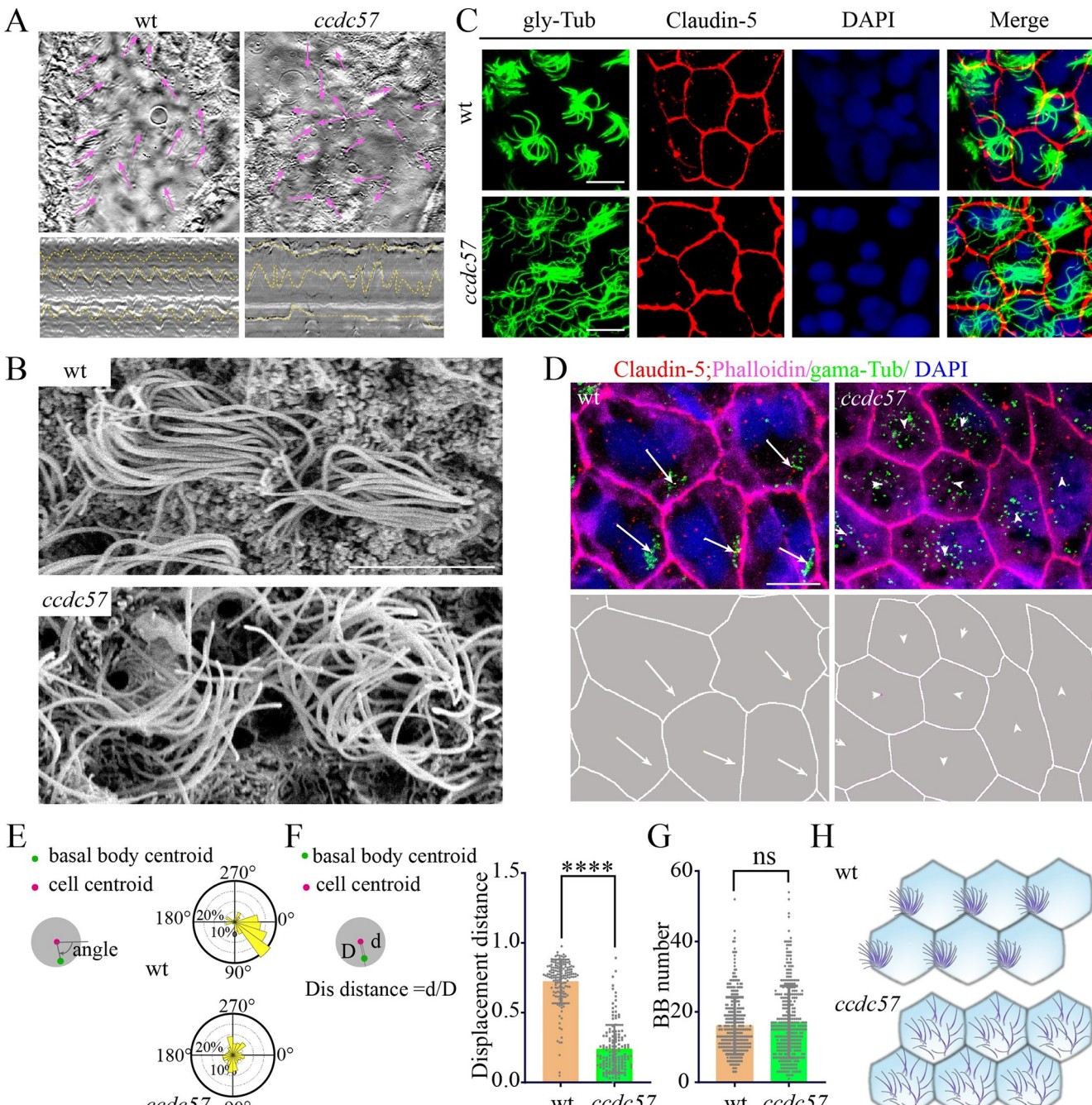

**Fig 3. Mutation of *ccdc57* results in planar polarity defects in ependymal cells.** (**A**) Still images showing the beating direction of motile cilia to drive fluid flow (pink arrow) in adult wild type and mutants. The representative kymographs of cilia movement were shown at the bottom. (**B**) Scanning electron microscopy showing the multicilia of the ependymal cells in wild type and *ccdc57* mutant as indicated. (**C**) Confocal images showing cilia of the ependymal cells in wild type and *ccdc57* mutant as indicated. Cilia were labeled with anti-mono-glycylated tubulin in green and the tight junction were stained with Claudin-5 antibody in red. Nuclei were counterstained with DAPI. (**D**) Confocal images showing the basal body distribution in the ependymal cells of wild type and *ccdc57* mutant as indicated. The basal bodies were labeled with anti-γ tubulin in green and cytoskeleton was stained with phalloidin in purple. Tight junctions were stained with Claudin-5 antibody in red. Nuclei were counterstained with DAPI. Arrows indicate the displacement of basal bodies from the center of the cells. The schematic diagram of the displacement of basal bodies were shown at the bottom. (**E**) Angular distribution of the basal bodies in ependymal cells from wild type and *ccdc57* mutant. (**F**) Statistical analysis showing relative displacement distance of basal bodies from center of the cell. (**G**) Statistical analysis showing the number of basal bodies in each ependymal cell of wild type and *ccdc57* mutant. BB, basal body. (**H**) Model illustrating the distribution of multicilia on ependymal cells of wild type and *ccdc57* mutant. All the data analyzed in this figure are generated from 3- to 4-months-old adult zebrafish. Scale bars: 5 μm in panels B and D; 7.5 μm in panel C. The data underlying the graphs shown in the figure can be found in S1 Data.

further confirmed via immunostaining with anti-glycylated tubulin antibody to visualize cilia (Fig 3C).

Cilia are anchored to the surface of ependymal cells through basal bodies. We therefore investigated basal body localization in wild type and *ccdc57* mutant ependymal cells. In wild type zebrafish, basal bodies appeared localized to the same side of each ependymal cell (Fig 3D and 3E). In contrast, basal bodies were randomly distributed in *ccdc57* mutants, with many localized to the center region of ependymal cells, indicating polarity defects in the absence of Ccdc57 (Fig 3D–3F). Notably, the number of basal bodies was comparable between mutant and wild type ependymal cells (Fig 3G). Next, we characterized the basal bodies of ependymal cells inside the ChP. In wild type zebrafish, multiciliated ependymal cells mainly localized to the center of the folds of the ChP (S5A and S5B Fig and S5 and S6 Movies). Of note, wild type ChP ependymal cells also exhibited planar polarity in basal body distribution (S5C Fig), and this planar polarity was absent in *ccdc57* mutant ChP ependymal cells (S5C–S5E Fig). Together, these data suggested that loss of Ccdc57 resulted in abnormal distribution of multicilia in the ependymal cells, together with basal body planar polarity defects (Fig 3H).

## Cell polarity defects of *ccdc57* mutant ependymal cells

We next asked whether ependymal cell polarity was affected in the absence of Ccdc57. Since basal bodies rely on the cytoskeleton microtubule network to localize cilia, the misplacement of basal bodies in *ccdc57* mutants may be due to abnormal organization of cytoskeletal microtubules. We therefore examined the cytoskeleton using alpha tubulin antibody staining and found that the microtubule skeleton appeared disorganized in *ccdc57* mutants as compared to the highly polarized microtubule network in wild type cells (S6A and S6B Fig). The position of mitochondria also depends on the microtubule network, we further analyzed the distribution of ependymal cell mitochondria using Tom20 staining. Our results showed that *ccdc57* mutant ependymal cells displayed abnormal mitochondria distribution patterns as compared with those of wild type cells (S6C and S6D Fig).

We next examined the shape of ependymal cells via Claudin-5 antibody-labeled cellular tight junctions. At 3 months postfertilization (mpf), wild type zebrafish ependymal cells appeared symmetrically distributed throughout the ependymal layer, with narrow and elongated cells located in the center and larger round cells located laterally (Fig 4A and 4B). Moreover, the elongated cell axes were parallel to the anterior–posterior axis, indicating polarity of these cells (Fig 4C). Intriguingly, the symmetrical distribution pattern of ependymal cells was also found to be established in 17 dpf wild type zebrafish larvae (Fig 4D–4F). In contrast, polarized ependymal epithelial cell organization was severely disrupted in 3 mpf *ccdc57* mutants (Fig 4B and 4C). At 17 dpf when spine curvatures are first apparent, *ccdc57* mutants also exhibited abnormal-shaped ependymal cells in the central region, suggesting an earlier defect of ependymal cell polarity (Fig 4E and 4F). Notably, since multiple motile cilia formation does not occur until 1 mpf (Fig 4G) [38], these results suggested that Ccdc57 participates in the regulation of ependymal cell polarity in early zebrafish development, as well as in later developmental basal body positioning.

## Ccdc57 orchestrates the synchronized beating of motile cilia in the spinal canal

During larval zebrafish development, ependymal cells are derived from radial glial cells, which also are involved in establishing their translational cell polarity [31]. Therefore, we next asked whether radial glial cilia were defective in the absence of Ccdc57. At larvae stages, spinal canal ependymal cells are a specialized type of radial glia harboring primary motile cilia that drive

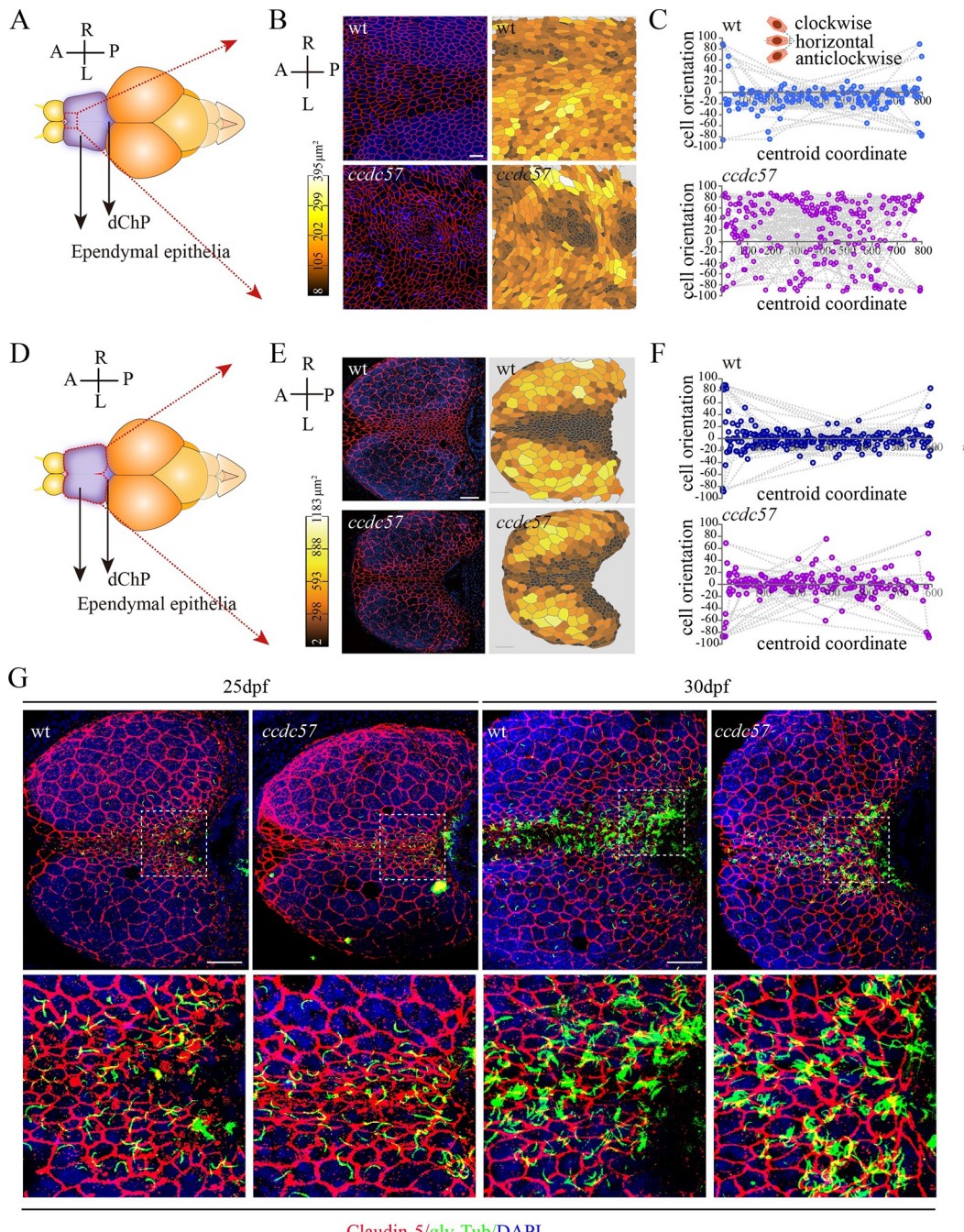

**Fig 4. Loss of Ccdc57 results in ependymal cell polarity defects.** (**A**, **B**) Distribution pattern of ependymal cells in 3-months-old wild type and *ccdc57* mutant as indicated with tight junction marker Claudin-5 staining. Nuclei were counterstained with DAPI in blue. The displayed region of ependymal epithelia was indicated in panel A. Heatmap showing the relative size of each ependymal cells as displayed in the confocal images. (**C**) Statistical analysis showing orientation of the midline ependymal cells along the anterior–posterior (A–P) axis in wild type and *ccdc57* mutant. The angles between the longer axis of each midline ependymal cells and the A–P axis were used to evaluate cell polarity with 0 degree indicating horizontal cell. (**D**, **E**) Distribution pattern of ependymal cells in the brains of 17 dpf wild type and *ccdc57* mutant. (**F**) Statistical analysis showing orientation of the midline ependymal cells in wild type and mutant as indicated. (**G**) Confocal images showing the ependymal layer labeled with Claudin-5 (red) and glycylated tubulin (green) antibodies in 25 and 30 dpf wt and *ccdc57* mutants. Enlarged views of the boxed regions are displayed in the bottom. Nuclei were counterstained with DAPI. Scale bars: 25 μm in panel B; 50 μm in panel E; 50 μm in panel G. The data underlying the graphs shown in the figure can be found in S1 Data.

CSF flow. In *ccdc57* mutants, the number and length of spinal canal cilia were comparable to those of wild type larvae (S2E and S2F Fig). Further examination of cilia localization by immunostaining with anti γ-tubulin antibody showed that both wild type and *ccdc57* mutant cilia were localized to the posterior apical surface of each radial glia cell in the spinal canal, with no apparent differences in basal body localization (Fig 5A and 5B).

We next examined the beating pattern of these spinal canal cilia using high speed video microscopy. By imaging cilia motility in the caudal central canal of 5 dpf wild type larvae, we found that all of the cilia beat in a similar manner, and a synchronized beating wave was easily observed throughout the field of view (Fig 5C and S7 Movie). However, the orchestrated ciliary beating was disrupted in *ccdc57* mutant spinal cords, with individual cilia beating independently of one another (Fig 5C and S8 Movie). Of note, the beating frequency of individual cilia was comparable between *ccdc57* mutant and wild type larvae (Fig 5D). Ciliary beating angle measurements demonstrated that the beating angle increased significantly in *ccdc57* mutants as compared to wild type siblings (Fig 5E). We further used the bisector of each angle to evaluate the tilting direction of motile cilia and found that the tilting angles were significantly increased in *ccdc57* mutants (cat. 72.11˚ ± 14.16˚ in wild type versus 85.52˚ ± 5.06˚ in mutants) (Fig 5F and 5G).

To further explore the beating defects of spinal canal cilia, we performed fluorescent bead tracing experiments at different stages of zebrafish development. By monitoring the movement of fluorescent beads injected into the central canal, we observed bidirectional CSF flow, as shown previously, in the central canal of wild type larvae at both 30 hpf and 3 dpf [39,40] (S9 and S11 Movies). The bidirectional CSF flow was largely maintained in *ccdc57* mutants, although residential circular particle movements were also observed (S10 and S12 Movies). Strikingly, we noticed that the injected fluorescent beads were able to be transported to the end of the central canal by 6 hours postinjection (hpi) in the majority of wild type larvae, while none of the *ccdc57* mutants contained fluorescent beads in the caudal spinal canal, and the fluorescent bead transport distances were significantly reduced (S7 Fig). The bead transport defects were observed in *ccdc57* mutants at all stages analyzed (3 dpf, 5 dpf, and 17 dpf) (S7 Fig). Together, these data suggest that Ccdc57 orchestrates the synchronized beating of spinal canal motile cilia, whose deficiency leads to abnormal CSF-flow.

## Localization of core PCP components in *ccdc57* mutants

The posterior tilting of spinal canal motile cilia is mainly regulated by the PCP pathway [41]. To further characterize this phenomenon, we examined the localization of two major PCP pathway components, Prickle and Dishevelled 1 (Dvl1), in wild type and *ccdc57* mutant fish. Similar to their basal bodies, Dvl1 appeared localized to the apical posterior region of each radial glia cell (Fig 5H), with Dvl1-positive vesicles localized to the region surrounding the basal bodies (Fig 5H). Noticeably, the Dvl1 distribution angles were comparable between *ccdc57* mutants and wild type control embryos (Fig 5I and 5J). The localization of Prickle was opposite to that of Dvl1 vesicles, in the anterior apical surface, as demonstrated by GFP-Prickle labeling. The Prickle localization was similar in wild type and *ccdc57* mutants (Fig 5K). These data suggested that localization of PCP components in larval zebrafish was not affected by the absence of Ccdc57. Intriguingly, in adult zebrafish, Dvl1 expressing vesicles appeared localized to one side of mature ependymal cells, corresponding to the location where multicilia formed (Fig 5L). In contrast, in adult *ccdc57* mutants, Dvl1 expressing vesicles appeared dispersed randomly throughout the entire cell, clearly indicating a late onset cell polarity defect in *ccdc57* mutants (Fig 5L).

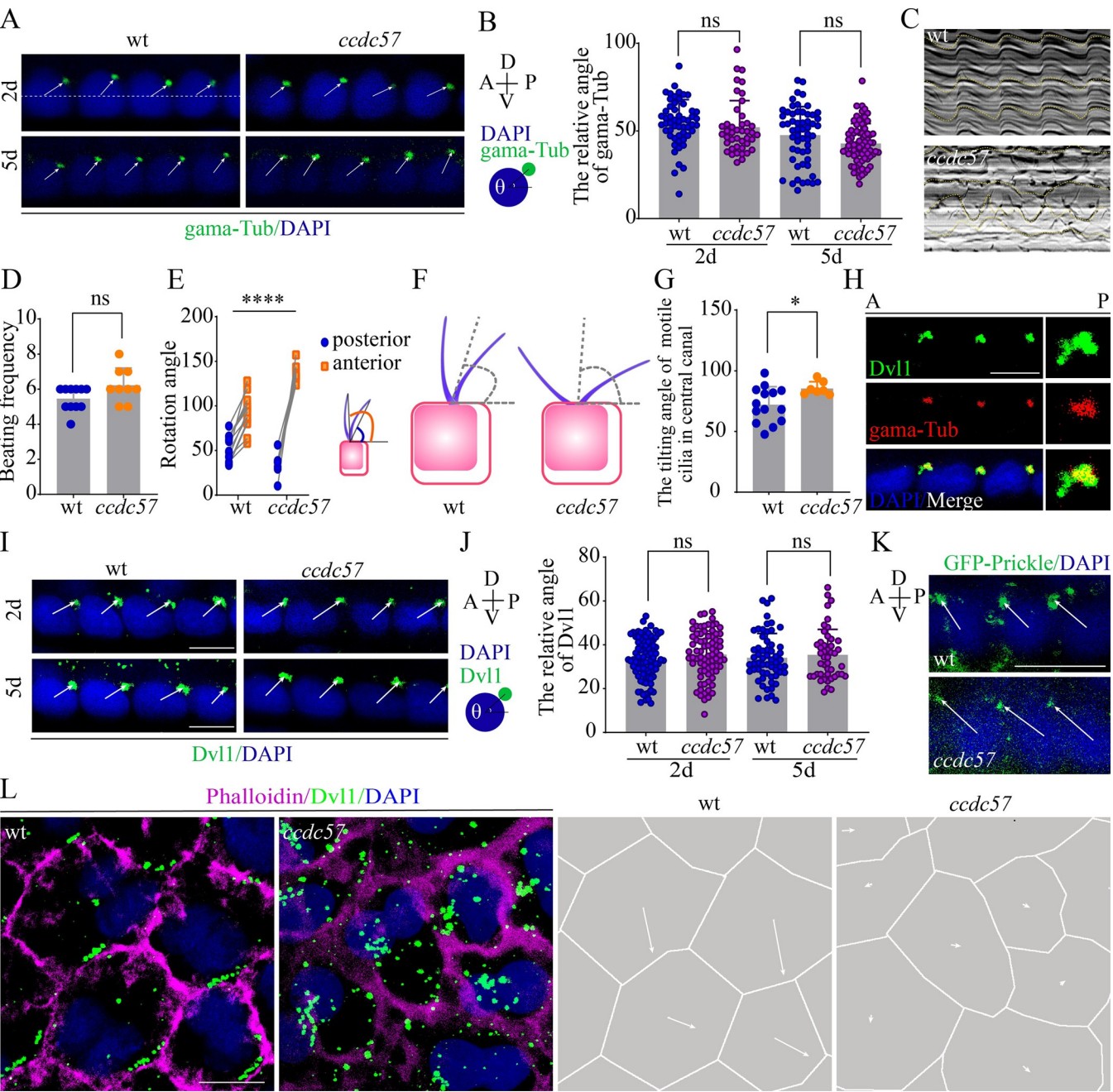

**Fig 5. Ccdc57 orchestrates synchronized beating of motile cilia in the central canal.** (**A**) Confocal images showing the localization of basal bodies in the floor plate cells at 2 dpf and 5 dpf. Basal bodies were stained with anti-γ tubulin (gama-Tub, green) antibody, and nuclei were counterstained with DAPI in blue. The white dotted line connects the center of cells. (**B**) Statistical analysis showing the angles of basal bodies with the anterior–posterior (A–P) axis as illustrated in the diagram. (**C**) Still images showing kymographs of floor plate cilia movement in 5 dpf of wild type and *ccdc57* mutant. (**D**) Beating frequency of floor plate cilia in wild type and *ccdc57* mutants. (**E**) Statistical analysis showing the rotation angles of cilia in floor plate. The anterior and posterior angles were measured as illustrated in the diagram. (**F, G**) Relative tilting directions of floor plate cilia in wild type and *ccdc57* mutants. The tilting direction was evaluated by the bisector of each angles. (**H**) Confocal images showing relative localization of Dvl protein (green) and basal body (gama-Tub, red) in floor plate cells. (**I, J**) Subcellular localization of Dvl protein (green) in the floor plate ependymal cells. The statistical results were shown in panel J. (**K**) Confocal images showing the localization of GFP-Prickle (green) in floor plate ependymal cells as indicated. (**L**) Confocal images showing the distribution of Dvl protein (green) on ependymal cells of adult zebrafish. The displacement distance of Dvl vesicles was also shown on the right. Scale bars: 5 μm in panel A; 5 μm in panels H and I; 10 μm in panel K; 7.5 μm in panel L. The data underlying the graphs shown in the figure can be found in S1 Data.

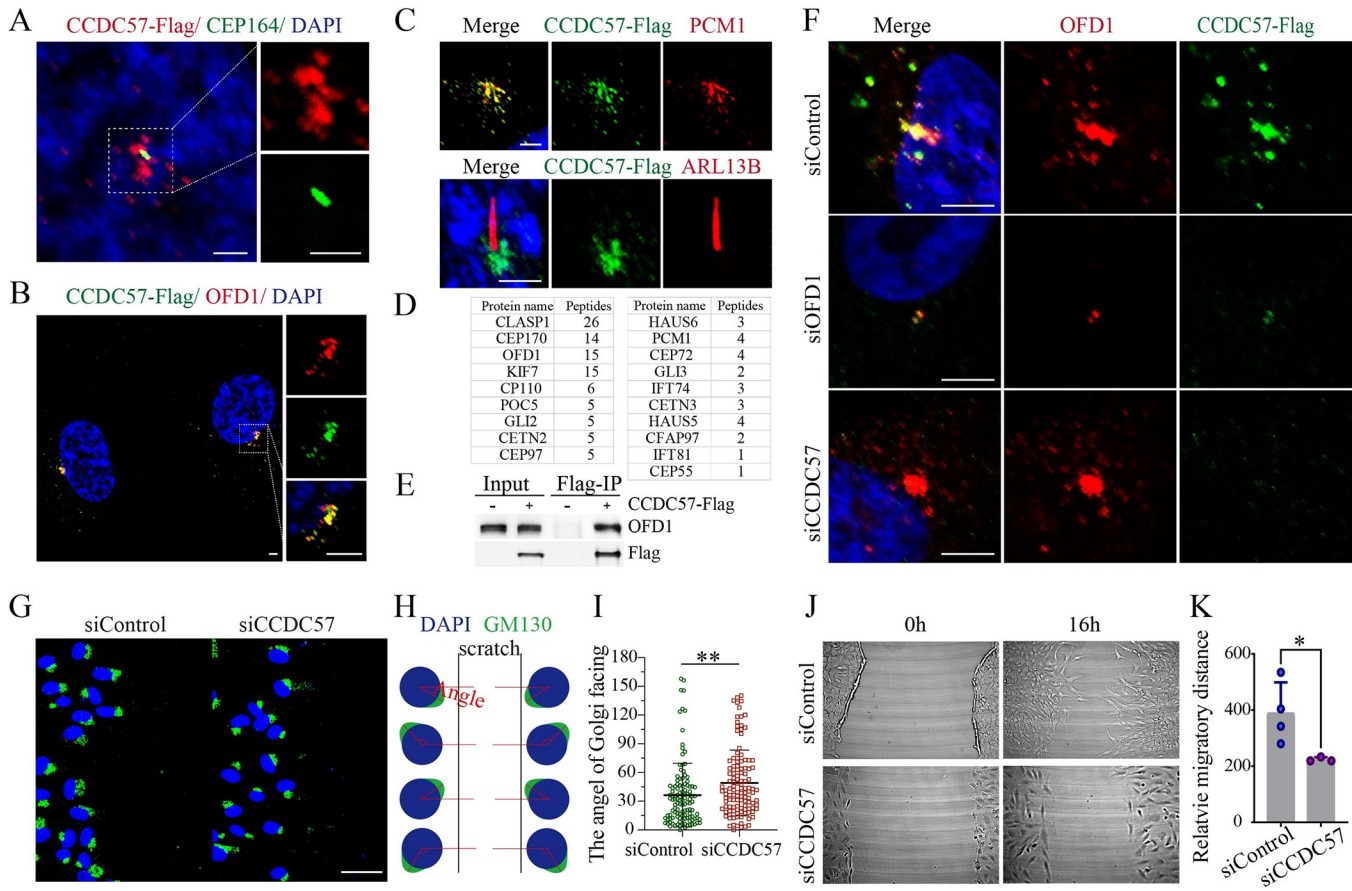

**Fig 6. CCDC57 is a centrosomal satellite protein required for cell polarity.** (**A-C**) Confocal images showing the relative localization of CCDC57 with OFD1, PCM1, CEP164, and ARL13B in RPE-1 cells. (**D**) Mass spectrometry analysis of the CCDC57 interacting proteins. (**E**) Immunoprecipitation results showing the interaction between CCDC57 and OFD1. (**F**) Confocal images showing the localization of CCDC57 and OFD1 in siRNA knockdown RPE-1 cells. (**G**) Scratch-wound assay showing the polarized localization of Golgi (GM130, green) during directional cell migration in control and CCDC57 siRNA-treated cells. (**H**) Model illustrating the statistical analysis of cell polarity by Golgi position. (**I**) Dot plots showing the angles of Golgi facing the migration edge in control and CCDC57 siRNA-treated cells. (**J**, **K**) Images showing the cell migration state and statistical analysis of the migration distance. In all panels, nuclei were counterstained with DAPI in blue. Scale bars: 2.5 μm in panel A; 5 μm in panel B; 3 μm in panel C; 5 μm in panel F; 50 μm in panel G. The data underlying the graphs shown in the figure can be found in S1 Data.

## *CCDC57* encodes a centrosomal satellite protein required for cell polarity

To reveal the mechanisms of Ccdc57 regulation of basal body positioning, we further examined the subcellular localization of CCDC57 in RPE-1 cells. Similar to previous reports, CCDC57 mainly localized to centriolar satellites in RPE-1 cells [42]. Flag-tagged CCDC57 protein colocalized with centriolar satellite proteins OFD1 and PCM1 and did not localize to cilia (ARL13B) or to the distal appendage of the basal body (CEP164) (Fig 6A–6C). Mass spectrometry analysis of pulled-down proteins using Anti-FLAG conjugated beads showed that CCDC57 interacted with multiple centrosomal proteins, including CEP170, OFD1, and CP110 (Fig 6D), and we further validated the CCDC57-OFD1 interactions via immunoprecipitation (Fig 6E). Moreover, siRNA knockdown of OFD1 eliminated localization of CCDC57 to the centriolar satellite. In contrast, siRNA knockdown of *CCDC57* gene expression had no effect on centrosomal localization of OFD1(Fig 6F).

During migration, cells can establish a front–rear polarity characterized by the polarized distribution of Golgi and centrosomes in the leading edge [43]. We therefore further tested the

role of CCDC57 in establishing cell polarity in RPE-1 cells by wound-scratch assay. After scratching, the microtubule cytoskeleton of leading edge RPE-1 cells appeared polarized, with the Golgi facing toward the scratched space to direct cell migration. In control siRNA-treated RPE-1 cells, the majority of leading edge cells contained Golgi apparatus located within 60 degrees of the direction of migration relative to the nucleus (Fig 6G–6I). In contrast, the polarized orientation of Golgi apparatus was significantly compromised in *CCDC57* siRNA-treated RPE-1 cells, as demonstrated by significantly increased orientation angles (Fig 6G–6I). Furthermore, migration distance was also reduced in *CCDC57* siRNA-treated cells as compared to control siRNA-treated cells (Fig 6J and 6K). Together, these in vitro studies further confirmed the role of the centrosomal protein, CCDC57, in establishing cell polarity.

## Abnormal RF assembly and urotensin expression in *ccdc57* mutant larvae

We next sought to identify the causes of the body curvature observed in *ccdc57* mutants. Epinephrine signals are essential for urotensin expression and body straightening [8]. In line with this, epinephrine treatment was also able to rescue body curvature in *ccdc57* mutant embryos (S8 Fig). The RF is essential for body axis straightening through transferring the epinephrine signals to the CSF-cNs [20,21]. We found that wheat germ agglutinin (WGA) can be used to label and image the RF (S9A–S9C Fig). In various ciliary mutants, WGA-labeled RF appeared either discontinuous or absent (S9D–S9G Fig). Similarly, WGA staining of the RF was also diminished or absent in *ccdc57* mutant larvae (Fig 7A). Interestingly, body curvature severity closely correlated with the severity of RF assembly defects in *ccdc57* or *kif3a* mutants (S9H–S9M Fig). Next, we examined the expression of urotensins in *ccdc57* mutant larvae using whole-mount in situ hybridization (WISH) analysis. While *urp1* expression appeared relatively

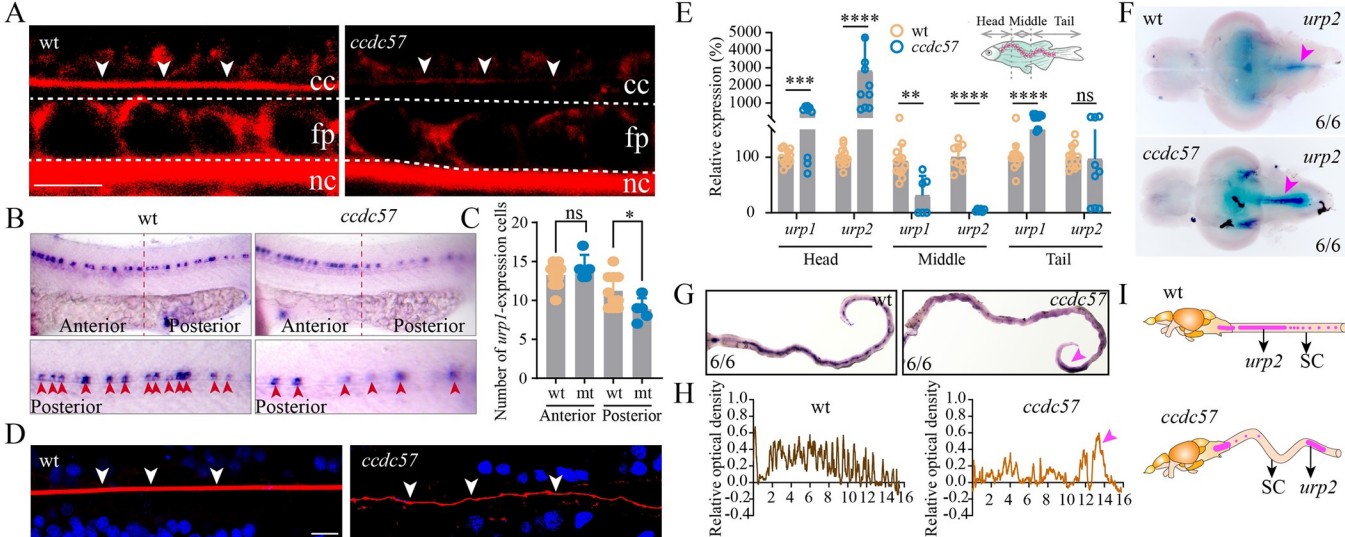

**Fig 7. Mutation of *ccdc57* results in RF defects and abnormal expression of Urotensins in the spinal cord.** (**A**) Representative images of Reissner fiber (RF, white arrows) in 2 dpf wild type and *ccdc57* mutant larvae. cc, central canal; nc, notochord; fp, floor plate. (**B**) Whole-mount in situ hybridization results showing the expression of *urp1* in 24 hpf control and *ccdc57* mutant larvae as indicated. The enlarged views of the staining in the posterior region are shown in the bottom. (**C**) Statistical analysis showing the number of *urp1*-expressing cells in the anterior and posterior part of the trunk. (**D**) Confocal images showing RF (white arrow) in wild type and *ccdc57* adult mutants. RF was stained with wheat germ agglutinin (WGA, red), and nuclei were counterstained with DAPI. (**E**) qPCR analysis showing the expression of urotensin genes (*urp1* and *urp2*) in different parts of the adult trunk as illustrated in the diagram. (**F**) In situ hybridization results showing the expression of *urp2* in the brains of adult wild type and *ccdc57* mutant as indicated. Arrows point to the expression of *urp2* in the posterior part of the brain. (**G**) In situ hybridization results showing the expression of *urp2* in the spinal cord of wild type and *ccdc57* mutant as indicated. The purple arrowhead indicates the sites of enriched *urp2* expression. (**H**) The line graphs showing the relative optical density of the expression of *urp2* in wild type and *ccdc57* mutants. (**I**) Model illustrating the distribution of *urp2* in wild type and *ccdc57* mutant. Scale bars: 7.5 µm in panel A; 10 µm in panel D. The data underlying the graphs shown in the figure can be found in S1 Data.

normal in the anterior trunk of *ccdc57* mutant larvae, the number of *urp1*-expressing cells was markedly decreased in the posterior trunk (Fig 7B and 7C). Together, these data suggested that Ccdc57 deficiency interrupts assembly of the RF and down-regulates the expression of urotensin genes. The fact that differences in urotensin gene expression mainly occurred in the posterior trunk may explain why *ccdc57* mutant larvae exhibit only mild body curvature.

## Ectopic accumulation of Urotensin neuropeptides in *ccdc57* adult mutants

Next, we attempted to discover the relationship between the development of scoliosis and CSF flow defects. First, we investigated the assembly of the RF in wild type and *ccdc57* mutants, based on the previously characterized roles for the RF in regulating body axis development [10,19,22]. In wild type adults, the RF appeared thick and straight (Fig 7D). In contrast, the RF appeared much thinner, discontinuous, and/or absent in adult *ccdc57* mutants (Fig 7D). Of note, the severity of RF assembly defects correlated with the level of scoliosis in the mutants (S10A and S10B Fig). Next, we focused on the expression of urotensin neuropeptides. Intriguingly, we found that from a lateral view, adult *ccdc57* mutant spines always contained a strong dorsal bending in the anterior trunk and a second dorsal bending in the tail region (Fig 1B and 1E). To better characterize this feature, we dissected adult wild type and *ccdc57* mutant trunks into three segments (Head, Middle, and Tail) and isolated total RNA from each segment (Fig 7E). Unexpectedly, qPCR results showed that the expression of *urp1* and *urp2*, the major urotensins regulated by CSF signaling, was increased over approximately 10 times in the Head segments of *ccdc57* mutants as compared with those of wild type control siblings (Fig 7E). In contrast, Middle trunk segments displayed lower expression levels of these genes in *ccdc57* mutants, while urotensin expression again appeared up-regulated in Tail segments of *ccdc57* mutant adult spines (Fig 7E). To further validate these results, we performed ISH analysis on dissected spinal cords for the expression of *urp2*, one of the major urotensin genes expressed at later stages. Our results showed that the expression of *urp2* was dramatically increased in the brainstem region of *ccdc57* mutants (Fig 7F). In contrast, *urp2* expression was virtually absent in the middle part of the spine, with some *urp2* expression observed in the tail region (Fig 7G and 7H). To further examine whether *urp2* expression in the tail region correlated with the second dorsal bending of the spine, we dissected the spinal cord at the second bending site and further cut it at the apex into anterior and posterior fragments (S10C Fig). Interestingly, compared with wild type control siblings, *urp2* gene expression was up-regulated in the posterior fragment of the dissected spinal cord, while *urp2* was not detected in the anterior fragment (S10C Fig). Together, these results suggest an interesting relationship between *urp2* expression and spinal curvature (Fig 7I).

## Correlation between urotensin expression and spine curvature in ciliary mutants

To further gauge the relationship between urotensin expression and spinal curvature, we evaluated spine curvature phenotypes in several zebrafish scoliosis mutants. Both *tmem67* and *ofd1* mutants displayed scoliosis. Micro-CT results showed that these mutants also displayed initial dorsal bending in the anterior spine (Fig 8A). The observed scoliosis in *ofd1* and *ccdc57* mutants further demonstrated the interactions between Ofd1 and Ccdc57 that were revealed by our IP pulldown experiments (Fig 6E). Of note, qPCR results confirmed the enhanced expression of *urp1* and *urp2* in the anterior spines of *ccdc57*, *tmem67*, and *ofd1* mutants (Fig 8B).

We have previously shown that mutation of *uts2r3*, the major Urotensin receptor, leads to scoliosis. Noticeably, the spinal curvature of *uts2r3* mutants was clearly different from that of ciliary mutants. The anterior dorsal bending phenotype appeared relatively minor in *uts2r3*

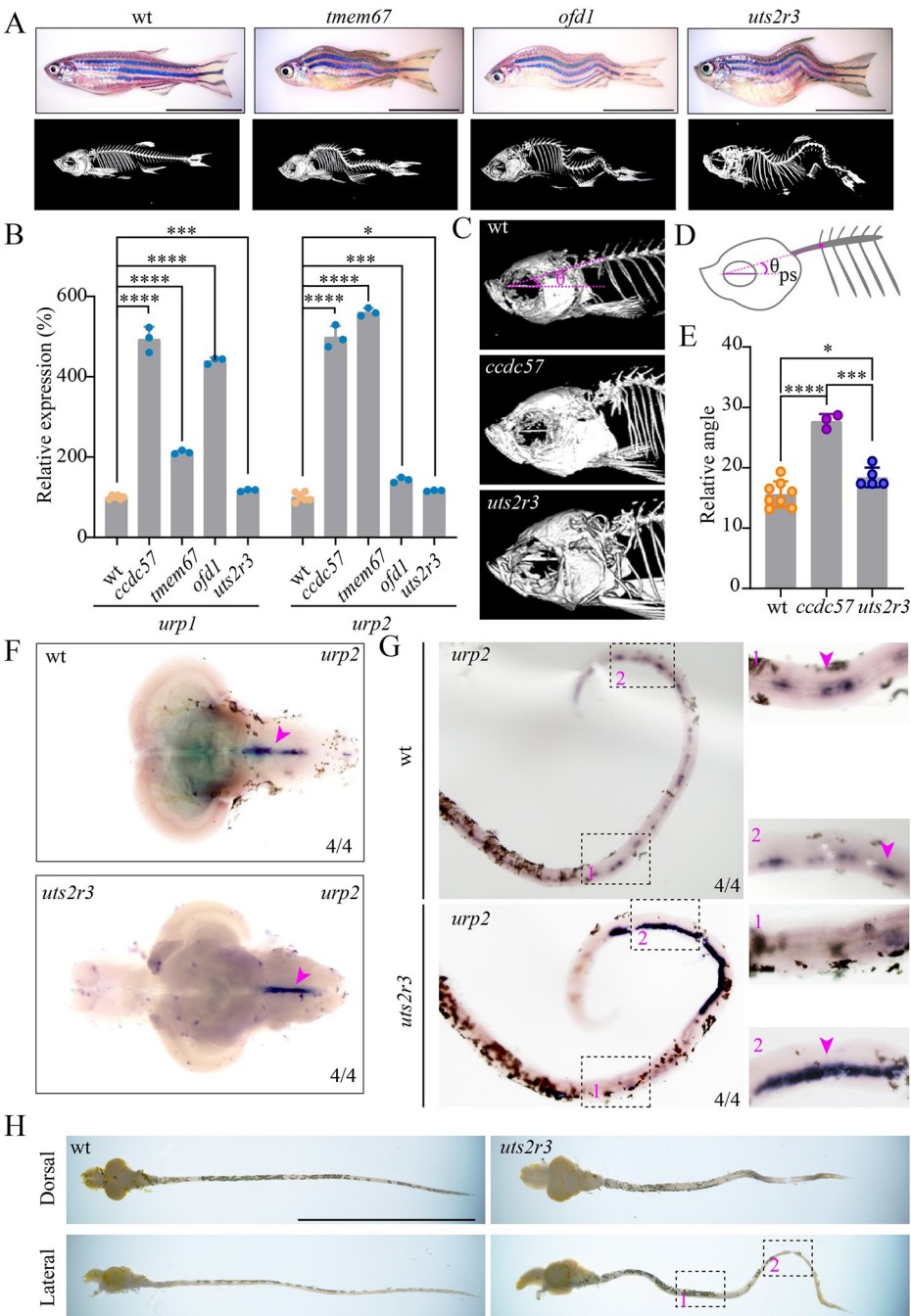

**Fig 8. Ectopic accumulation of urotensin neuropeptides is associated with spine curvature in zebrafish ciliary mutants.** (**A**) Representative images of wild type and scoliosis zebrafish mutants. Micro-CT images are shown on the bottom. (**B**) qPCR analysis showing the expression of urotensin genes in the heads of wild type and scoliosis mutants. (**C**) Enlarged views of the head regions of wild type, *ccdc57*, and *uts2r3* mutants. (**D, E**) Statistical analysis of the dorsal curvature angles in different mutants as indicated. The angles were measured between the direction of the parasphenoid bone and the Weberian vertebrae as illustrated in the diagram (**D**). (**F**) In situ hybridization results showing the expression of *urp2* in the brains of wild type and *uts2r3* mutant as indicated. (**G**) Expression of *urp2* in the spinal cords of wild type and *uts2r3* mutant. The strongly increased expression of the *urp2* in the tail region was shown on the enlarged views. (**H**) Dissected spinal cords from wild type and *uts2r3* mutants as indicated. The two boxed regions correspond to those in panel G. Scale bars: 1 cm in panels A and H. The data underlying the graphs shown in the figure can be found in S1 Data.

mutants, while all *uts2r3* mutants bent to the ventral side first, and displayed strong dorsal bending in the posterior portion of the trunk (Fig 8A). To better characterize spinal bending, we measured the angle between the parasphenoid bone and the Weberian vertebrae orientation in wild type, *ccdc57*, and *uts2r3* mutants (Fig 8C). The measured angles of *ccdc57* mutants were significantly larger than those in control and *uts2r3* mutants (Fig 8C–8E). Consistent with these results, *urp2* expression levels were dramatically increased in the anterior spines of *ccdc57* mutants, while only a slight increase was observed in *uts2r3* mutants (Fig 8B), as validated via WISH assay (Fig 8F). Noticeably, the location of increased *urp2* expression in *uts2r3* mutant spines also correlated with the observed second bending in *uts2r3* mutant spines (Fig 8G and 8H). Together, these data provide strong evidence that spinal bending closely correlates with the activation of Urotensin signaling.

### Abnormal urotensin signals in idiopathic scoliosis patients

Finally, we sought to investigate whether urotensin signaling is also involved in the regulation of spinal curvature in human scoliosis patients. Although it is difficult to obtain scoliosis patient spinal cord tissue for analysis, we were able to collect bilateral paravertebral muscle tissue from scoliosis patients during surgery. We then compared Urotensin signaling pathway gene expression in paravertebral muscle tissue harvested from the convex and concave sides of spinal curvature sites (Fig 9A). Strikingly, although *UTS2* expression appeared similar between convex and concave muscle tissue locations, we observed a remarkably asymmetric expression of *UTS2R* in bilateral paravertebral muscles of AIS patients ($n$ = 46) (Fig 9B and 9C). The expression of *UTS2R* was significantly higher in convex side as compared to concave side muscle tissue (Fig 9B). According to the ratio of *UTS2R* expression in the convex versus the concave (convex/concave), we classified these AIS patients into two groups with the ratio of 2 as cutoff point. Strikingly, patients with >2-fold difference in expression ($n$ = 26) had remarkably severe curvature magnitude as compared to patients with <2-fold difference ($n$ = 20) (58.62 ± 9.48 degrees versus 51.35 ± 6.58 degrees) (Fig 9D and 9E). Thus, these data strongly support the hypothesis that abnormal Urotensin signaling may make a significant contribution to the severity of spinal curvature observed in AIS patients.

In summary, here we showed that Ccdc57 is a centrosomal protein required for the proper establishment of basal body and cilia polarity. Loss of Ccdc57 disrupted the planar polarity of ependymal cells, affected the polarity and beating pattern of cilia present in both radial glial and ependymal cells, and led to CSF flow defects. The CSF flow defects resulted in the up-regulation of Urotensin signals in the anterior portion of the spine, eventually leading to spinal dorsal bending and scoliosis.

## Discussion

Scoliosis is one of the most common diseases diagnosed in childhood or early adolescence, although the underlying causes remain largely unknown. Recent studies suggest that in addition to environmental factors, underlying genetic factors also contribute to the incidence of spinal curvatures. The coincidence of scoliosis and hydrocephalus in several human genetic disorders suggests a potential relationship between hydrocephalus and spinal curvature development. Similarly, hydrocephalus is constantly observed in zebrafish ciliary scoliosis mutants. Remarkably, the curvature of the spine is not apparent until approximately 3 weeks fertilization in virtually all zebrafish ciliary mutants, which points to an interesting question: Why does abnormal spine curvature develop at these stages? In this paper, we present data showing that deficiency of the centrosomal protein, Ccdc57, results in scoliosis in zebrafish. Our data suggest that reduced Ccdc57 expression results in ependymal cell polarity defects as early as 17

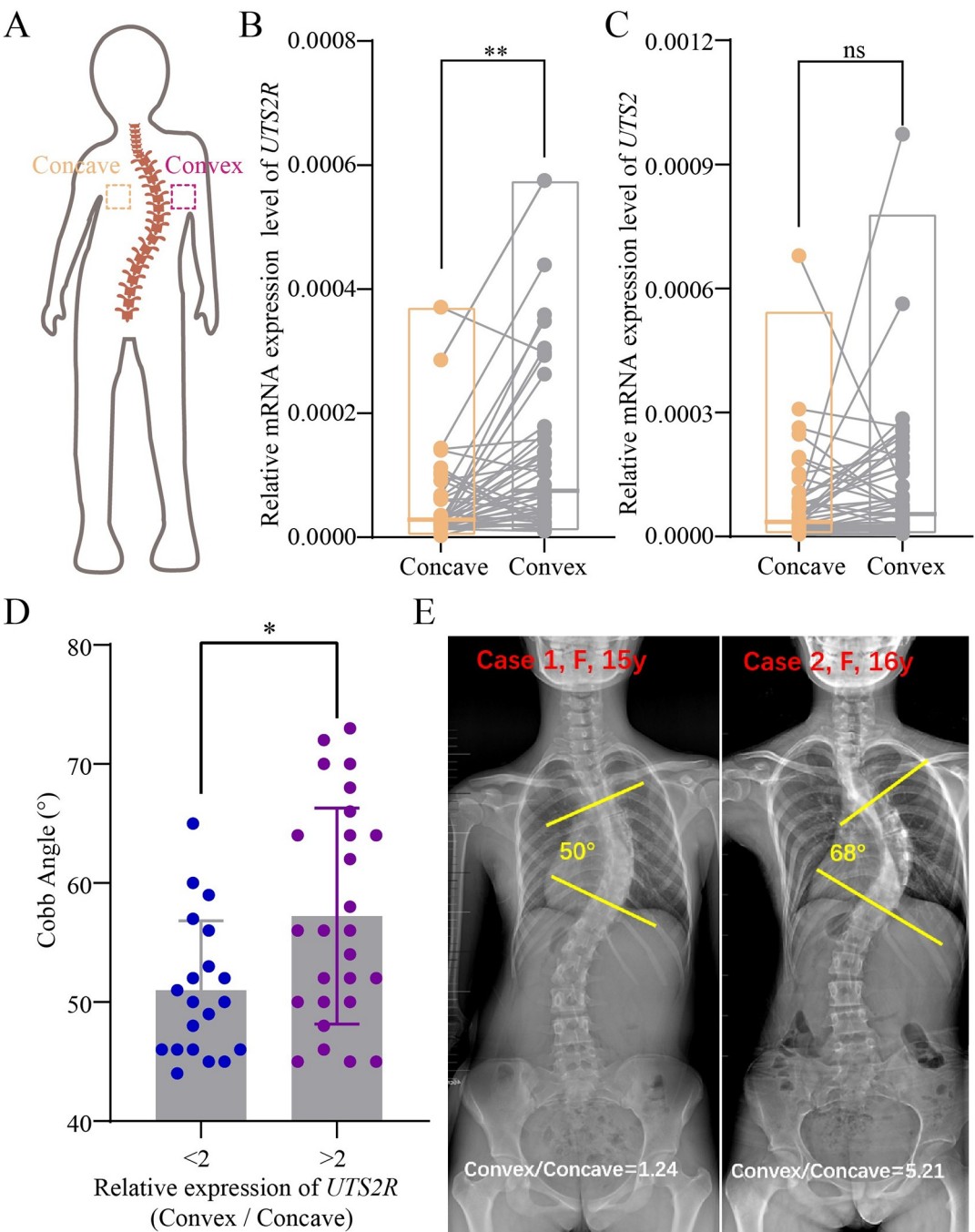

**Fig 9. Abnormal urotensin signals in adolescent idiopathic scoliosis (AIS) patients.** (**A**) Diagram indicating the position of bilateral paravertebral muscles harvested from AIS patients. (**B**, **C**) Relative expression of *UTS2R* and *UTS2* in bilateral paravertebral muscles of AIS patients ($n = 46$). (**D**) Distribution of Cobb angles within the two groups of AIS patients exhibiting a ratio of UTS2R expression on the convex versus concave sides greater ($>$) or less ($<$) than 2. (**E**) Representative X-ray images showing two AIS patients with convex/concave ratios greater (5.21) or less than 2 (1.24). Measurements of Cobb angles are also indicated. The data underlying the graphs shown in the figure can be found in S1 Data.

dpf, which, in turn, affect the coordinated beating of ependymal cell multicilia and CSF flow. Moreover, the subsequent accumulation of CSF causes up-regulation of the Urotensin expression in the anterior spine, which contributes to the initial spine curvature formation.

In mouse, multiciliated ependymal cells display planar polarity in three distinct levels: the direction of cilia beating known as rotational polarity; the displacement of cilia via the basal body positioning known as translational polarity; and the alignment of ciliary basal bodies at the intercellular level known as tissue polarity [44]. Similarly, our data suggest that multiciliated ependymal cells also displayed these three types of polarities in zebrafish. In wild type zebrafish, most basal bodies appeared localized to the same side of individual ependymal cells. In *ccdc57* mutants, both basal body placement and coordinated cilia beating were disrupted in ependymal cells. Noticeably, synchronized cilia motility can occur at the cellular (each individual cilium) and intercellular (cilia bundle) levels, and both were affected in the absence of Ccdc57. The beating direction of cilia is closely related to the orientation of the basal foot, an accessory structure projecting from the side of the basal body. Within each ependymal cell, the basal feet are aligned towards the direction of cilia beating [45]. Considering the subcellular localization of CCDC57, it is conceivable that Ccdc57 deficiency may lead to misalignment of ependymal cell basal feet. Moreover, the rotational orientation of cilia is also regulated by several PCP-core modules including Frizzled, Van Gogh-like (Vangl1/2), and Prickle. PCP of ependymal cells is first established through the asymmetric localization of these core PCP components via a polarized microtubule network, which further organizes the distribution of ciliary basal bodies in a PCP-dependent manner [32,46–48]. Interestingly, while the distribution of the PCP proteins Dvl1 and Prickle was initially normal in the radial glial cells of larval mutants, adult *ccdc57* mutants showed disorganized basal body distribution, suggesting that Ccdc57 may function downstream of PCP components to regulate basal body positioning. Finally, *ccdc57* mutant ependymal cells displayed abnormal cell polarity characterized by microtubule polarity defects, which occurred before the maturation of multiciliated cells. These data suggest that organizational defects in the microtubule network were upstream of the observed abnormal basal body distribution in *ccdc57* mutants. Ccdc57 may regulate the polarity of the microtubule network via its microtubule binding domain [42]. Since the basal foot also participates in the regulation of the polarity of microtubule network [49,50], Ccdc57 may also control the basal body polarity to organize the microtubule network. In the absence of Ccdc57, the microtubule network polarity is defective, resulting in later defects of abnormal cilia beating and basal body positioning.

Defects in motile cilia of the brain ventricles are associated with hydrocephalus in humans, mice, and zebrafish. While *ccdc57* mutants displayed strong hydrocephalus and multiciliated cell defects, it is surprising that brain morphology appeared relatively normal in *gmnc* zebrafish mutants that lack multiciliated cells [38,51] (S11A–S11C Fig). Moreover, *gmnc* mutants did not exhibit scoliosis, suggesting that defective multicilia were not the prerequisite for CSF flow defects and the progression of scoliosis (S11D Fig). In contrast to *ccdc57* mutants, coordinated single motile ciliary beating was relatively normal in *gmnc* mutants (S11E Fig and S13 and S14 Movies), which, together with heartbeat and body movements, ensured sufficient circulation of CSF in the CNS [52]. In line with this, the RF also developed normally in *gmnc* mutants (S11F Fig). In adult *gmnc* mutants, although multicilia were not differentiated as demonstrated by the localization of basal bodies, the polarity displacement of primary motile cilia remained largely normal (S11G Fig). Moreover, the distribution pattern of ependymal cells also appeared relatively normal, as compared to the severely disorganized pattern observed in *ccdc57* mutants (S11H Fig). Therefore, it is likely that the single motile cilia in *gmnc* mutant ependymal cells are sufficient to drive CSF flow. In contrast, coordinated ciliary beating is essential to prevent the development of hydrocephalus. In *ccdc57* mutants, the PCP defects resulted in dampened directional CSF flow due to lack of coordinated beating of individual cilium in the ependymal cells. The CSF flow defects further resulted in CSF accumulation in the brain ventricles, leading to hydrocephalus. It is noteworthy that hydrocephalus forms from the

cumulative effects of defective CSF flow, as only a small number of *ccdc57* mutant larvae exhibited hydrocephalus at 17 dpf when spinal curvature was initially observed (S12A and S12B Fig). In contrast, the width of spinal canal was significantly increased in the mutants starting from as early as 3 dpf (S12C and S12D Fig), an earlier sign of CSF flow defects.

In summary, we return to our question: How does hydrocephalus result in scoliosis? We found that ependymal cell polarity defects in *ccdc57* mutants first occurred at approximately 17 days after fertilization, prior to the formation of differentiated multicilia. Interestingly, scoliosis was also first apparent at this developmental stage, suggesting a potential relationship between cell polarity defects and scoliosis. We have shown previously that the Urotensin signals govern zebrafish body axis straightening through activating its receptors located in dorsal muscle fiber cells [8]. Analyses of dissected antero-posterior portions of adult *ccdc57* mutant spines revealed that the expression of Urotensin neuropeptides was significantly up-regulated in the head region. In *ccdc57* mutants, CSF flow defects led to hydrocephalus and excess accumulation of CSF in the brain ventricles. The observed enhanced expression of *urp2* in the hindbrain may be caused by up-regulation of epinephrine signaling in the brain ventricles due to CSF accumulation. Interestingly, we found that all *ccdc57* mutants developed dorsal curvature in the anterior part of the spine. Since Urotensin signals can promote the contraction of dorsal muscle fibers, it is possible that the anterior-most, first dorsal curvature was the result of enhanced *urp2* expression in this region. In fact, almost all reported zebrafish ciliary scoliosis mutants displayed a first dorsal curvature phenotype [6,22,53,54]. In contrast, *uts2r3* mutants that lack urotensin receptors displayed minor dorsal bending in the anterior portion. Of note, the second dorsal bending of the *ccdc57* mutants observed in the posterior portion of the trunk also corresponded to enhanced *urp2* expression. It is likely that these secreted neuropeptides activate the contraction of muscle fibers locally and that the uneven distribution of Urotensin neuropeptides in *ccdc57* mutant results in unbalanced muscle contraction surrounding the spine, eventually leading to scoliosis.

Finally, data obtained from human AIS patients suggested that Urotensin signals were also differentially activated between the convex and concave sides of the spine in these patients. Interestingly, the observed up-regulation of *UTS2R* expression in AIS patients also occurred on the convex side of the spine, similar to the up-regulation of *urp2* observed in the dorsal bending sites of zebrafish mutant spines. It is noteworthy that we did not observe differential expression of urotensin neuropeptides between the convex and concave spinal muscle tissues of AIS patients. We believe this result may be due to the fact that AIS patient tissues were collected from paravertebral muscles that were enriched for the expression of UTS2R, but not for the neuropeptides, which are mainly secreted from the neurons.

Altogether, our data suggest that ependymal polarity defects are the earliest sign of scoliosis development in zebrafish ciliary mutants, and we provide one explanation for how scoliosis develops in zebrafish ciliary mutants, which may provide new insight into mechanisms regulating scoliosis in humans.

# Materials and methods

## Ethics statement

All zebrafish studies were conducted according to standard animal guidelines and approved by the Animal Care Committee of Tufts University and Ocean University of China. Human tissue collection was approved by the Ethics Committee of the Nanjing University Medical School Affiliated Nanjing Drum Tower Hospital, China (No. 2019-066-01).

## Zebrafish strains and mutants

All zebrafish strains were maintained at 14-hour light/10-hour dark cycle at 28.5°C. The $tft^{168N}$ mutant line was identified from an ENU-based mutagenesis screen conducted in the Yelick Laboratory, Tufts University. The chromosomal location of the $tft^{168N}$ allele was mapped by the Goessling Laboratory, Harvard-MIT Division of Health Sciences and Technology to an interval on Chromosome 12 containing 3 premature stop codon gene mutations, including a tgt/tga nucleotide substitution resulting in a predicted C/* mutation at amino acid 573 of *ccdc57*. Further analysis via quantitative real-time PCR (qRT-PCR) validated this homozygous mutation in phenotypic $tft^{168N}$ mutants. Further validation of the *ccdc57* mutation and scoliosis phenotype was performed via generation of CRISPR/Cas9 *ccdc57* mutants. Zebrafish mutants were generated using CRISPR/Cas9 method with the following target sequences: *ccdc57*: 5′-GGGAAGAGGTCAGTGAGCTT -3′, *ofd1*: 5′-TATCAGACCTTCAAGAGCCG-3′, *tmem67*:5′-GGCAAGTGTCAGTGTCCTGA-3′. Cilia mutants (*MZkif3a*, *ift74*, *ift88*, and *uts2r3*) and *Tg (urp1*: *GAL4; UAS*: *Kaede)* transgenic lines were the same as previously reported [8,9,55–57].

## Spinal cord and brain dissection

To dissect zebrafish spinal cords, humanely killed adult zebrafish were fixed in 4% of paraformaldehyde (PFA) overnight at 4°C. After washing 3 times with 1X PBS (phosphate buffered saline (PBS)) for 20 minutes each, the skin, muscle, and neural arches were carefully removed with tweezers, and the spinal cord was separated from the vertebral bodies. For brain dissection, zebrafish were first humanely killed using Tricaine in ice-cold water, then the skulls were removed with tweezers in 1X PBS to expose the brain. The dissected brain and spinal cord were further processed for WISH according to standard protocols.

## Alizarin red and Calcein staining

Adult zebrafish were fixed in 4% PFA at 4°C for 1 week. After a 1-hour wash with PBST (0.1% Tween20), fixed zebrafish were stained with 0.01% Alizarin red (Sigma) in 1% KOH for another week. Next, the samples were washed in PBST for 3 days (1 wash/day) at room temperature and further cleared with 0.5% trypsin digestion for 24 hours at room temperature. After multiple washes with PBST, the skin was manually removed and the skeleton was imaged on ZEISS stemi 508 microscope. For Calcein staining, live zebrafish were incubated in 0.2% Calcein (Sigma) (pH 7.5) for 15 minutes and then washed twice with system water. In vivo stained zebrafish were anesthetized with 0.01% tricaine methanesulfonate (MS222), mounted in 3% methyl cellulose, and imaged using a fluorescent Leica M165FC microscope.

## Micro-CT and vibratome sectioning of brain tissue

To visualize mineralized skeletal structures via Micro-CT, zebrafish were first anesthetized with 0.01% tricaine methanesulfonate (MS222). Micro-CT images were captured using a PerkinElmer Quantum GX2 microCT scanner. For histological analysis, the dissected brain was fixed in 4% PFA overnight at 4°C and then embedded in 3% low melting agarose. Transverse serial sections through the brain were collected using Leica vibratome vt1000s at a thickness of 90 μm each and imaged using Leica M165FC microscope.

## High-speed video microscopy

To record the motility of ependymal cell cilia, the brain was dissected from the adult zebrafish, and the ependymal cell layer was gently peeled away from the top of the telencephalon with

tweezers. The ependymal tissues were positioned on a glass cover slip containing 1X PBS and placed upside down in the center of a depression slide. Cilia movement was recorded using a 100X oil objective on a Leica Sp8 confocal microscope equipped with a high-speed camera (Motion-BLITZ EoSens mini1; Mikro-tron, Germany). Recordings of in situ cilia movement in the posterior portion of the spinal canal of 5 dpf larvae were conducted as previously described [58]. In all experiments, cilia movement was captured at a rate of 500 frames per second, and playback was set at 25 frames per second. Image processing was performed using ImageJ software (National Institutes of Health, Bethesda, MD, USA).

## Scanning electron microscopy

Ependymal epithelia were dissected from the brain, fixed in 2.5% glutaraldehyde overnight at 4˚C, then washed 3 times with 1X PBS. Next, samples were serially dehydrated to 100% ethanol and transferred into isoamyl acetate. After critical point drying, samples were sputter coated with gold–palladium alloy before imaging. Images were acquired using Hitachi-3400N scanning electron microscope (Hitachi, Tokyo, Japan).

## Immunofluorescence assay

Immunostaining of whole-mount larvae was performed using standard protocols [59]. For immunofluorescence of ependymal cells, the brain was first dissected from adult zebrafish and then fixed in Dent's fixative (80% methanol and 20% dimethylsulfoxide). Fixed brains were incubated with primary antibodies overnight at 4˚C, followed by washing 3 times with PBST at 4˚C, then incubated with secondary antibody overnight at 4˚C. Immunostained adult zebrafish ependymal cell layer tissues were carefully collected with tweezers for imaging. The ependymal cell tissues of zebrafish less than 1 month old were imaged directly and did not require dissection from the brain.

For RF staining, 48 hpf zebrafish larvae were fixed in Dent's fixative overnight at 4˚C, washed 3 times with 1 X PBS, then incubated with WGA dissolved in PBS (1:200) overnight at 4˚C. After 4 times wash with 1X PBS, the samples were imaged with Leica Sp8 confocal microscope. For staining in adult zebrafish, the spinal cord was first dissected, fixed in 4% PFA overnight at 4˚C, and further processed for WGA staining. After staining, the tissues surrounding the spinal cord were removed to expose the RF for imaging.

The following antibodies were used: anti-glycylated tubulin (1:500, EMD); anti-γ-tubulin (1:500, Sigma); anti-cldn5 (1:250, Thermo Fisher Scientific); anti-DVL1 (1:200, BiCell Scientific); anti-TOM20 (1:500, Abcam); anti-alpha-tubulin (1:500, Sigma); anti-polyglutamylated tubulin (1:500, Adipogen); and WGA (1:200, Invitrogen).

## Cell culture

HEK293T cells were cultured in Dulbecco's Modified Eagle Medium-high glucose (D6429, Sigma-Aldrich), and RPE-1 cell lines were cultured in Dulbecco's Modified Eagle Medium/ Nutrient Mixture F-12 (D8437, Sigma-Aldrich). Both media were supplemented with 10% fetal bovine serum (FSP500, ExCell Bio) and 100 IU/mL penicillin/streptomycin (15140122, Gibco).

## Plasmid and siRNA transfection

Transfections of HEK293T cells were performed using polyethyleneimine (PEI) (24765–2, Polysciences). Transfections of siRNA were performed using Lipofectamine RNAiMAX (Thermo Fisher, 13778150). All transfections were performed according to the manufacturer's

instructions. The final concentrations of control siRNA (D-001810-10, Dharmacon) and CCDC57 siRNA (L-021433-02, Dharmacon) were 30 nM.

## Lentiviruse package and infections

The pLV-Flag-CCDC57 plasmids were cotransfected with package plasmids psPAX2 (Addgene #12260) and pMD2.g (Addgene #12259) into HEK293 cells. Supernatant media containing lentivirus was filtered through a 0.45-μm membrane filter and added to plated RPE-1 cells in the presence of 5 μg/ml polybrene (sc-134220, Santa Cruze). After 48 hours, the infected cells were selected with 10 μg/mL puromycin for 14 days in vitro culture.

## Immunofluorescence microscopy of cultured cells

The methods used were as previously described [60]. Briefly, cells grown on coverslips were fixed with 4% PFA for 10 minutes at room temperature and then permeated by cold methanol for 10 minutes at −20˚C. After washing with PBS, the cells were incubated with the primary antibodies at room temperature for 1 hour. After washing twice with PBS, cells were incubated with secondary antibodies and 4′, 6-diamidino-2-phenylindole (DAPI) for 1 hour in dark. After washing 3 times with PBS, the coverslips were mounted with mounting buffer (S36963, Invitrogen). All antibodies were diluted in blocking buffer (1% BSA in PBS). Images were acquired using laser scanning confocal microscope (FV3000, Olympus) with a 40× oil-immersion objective. ImageJ software was used for quantification. For IF, the following antibodies were used: mouse anti-flag (1:1,000, Sigma-Aldrich); rabbit anti-PCM1(1:1,000, Cell Signaling Technology); rabbit anti-CEP164 (1:4,000, Proteintech); and OFD1 (1:5,000, homemade).

## Coimmunoprecipitation and mass spectrometry analysis

HEK293T cells transfected with pLenti-Flag-CCDC57 were lysed in lysis buffer (20 mM Tris–HCl (pH = 7.5), 150 mM NaCl, 1 mM EDTA, 1 mM NaF, 0.3% Triton X-100) with 1% protease inhibitor cocktail (K1007, APExBIO). After centrifugation at 15,000*g* for 10 minutes at 4˚C, the supernatant was incubated with FLAG M2 beads (A2220, Sigma-Aldrich) for 2 hours at 4˚C, followed by 3 washes with lysis buffer.

Proteins were eluted from FLAG M2 beads using 0.2 mg/mL FLAG peptides (F3290, Sigma-Aldrich), mixed with SDS sample buffer and resolved using SDS-PAGE. Finally, protein bands were excised from the SDS-PAGE gels, digested with trypsin, and the extracted peptides were analyzed using mass spectrometry.

## Western blot analysis

Proteins were separated by 10% SDS-PAGE and transferred to NT nitrocellulose membranes (66485, BIOTRACE). After blocking with 5% milk, membranes were incubated with anti-FLAG (1:5,000; F1804; Sigma-Aldrich) and anti-OFD1 (1:30,000) for 1 hour at room temperature. After washing with PBS 3 times, the membranes were incubated with secondary antibodies for 1 hour at room temperature. The ChemiDoc Touch Imaging System (Bio-RAD) and Odyssey Laser Imaging System (LI-COR) were used for image acquisition.

## Wound healing assays

RPE1 cells transfected with negative control or *CCDC57* siRNAs were seeded on glass coverslips in 12-well plates. When reaching 100% confluence, the culture medium was replaced with serum-free media. After 24-hour starvation, 1 mL pipette tips were used to generate scratches on the glass coverslips, and PBS was used to wash away the detached cells.

Subsequently, cells were cultured in serum-free medium for an additional 6 hours and then fixed with 4% PFA. Golgi apparatuses were stained with GM130 (1:500; M179-3, MBL). Cells with Golgi apparatuses located in the sector facing the wound were considered positive for migration. Three groups (>100 cells per group) were used for quantification. GraphPad Prism software was used for statistical analysis. Quantitative data were presented as mean ± standard deviation (SD), and Student $t$ tests were performed to determine statistical significance.

## Pharmaceutical treatments and whole-mount in situ hybridization

Epinephrine (Sigma) treatment was performed as previously described [8]. Briefly, mutant and control zebrafish embryos were incubated with epinephrine (10 mg/ml) from bud stage until analysis. Images were captured using a Leica M165FC microscope. WISH was performed according to standard protocols. The following primers were used for in situ hybridization: *urp1* forward: 5′-ACATTCTGGCTGTGGTTTG-3′, reverse: 5′-TGTATGGGGAAAACAAAGG-3′; *urp2* forward: 5′-CAGCCCAAATAACAGAGACAAGAG-3′, reverse: 5′- AGAGGGT-CAGTCGTGTTATTGAGG -3′.

## Quantitative PCR

The expression levels of urotensin genes were evaluated using qRT-PCR analysis. Guided by the curvature of the spines, scoliosis mutant trunks were cut into three segments to include the highest or lowest portions of each spinal curve. Total RNA was isolated from dissected larvae or adult tissues using Trizol (Takara). cDNA was synthesized using PrimeScript 1st strand cDNA Synthesis Kit (Takara). qPCR was performed on the Step One real-time PCR system (Thermo Scientific) using the Eva-Green Master Mix (ABM). The following primers were used for PCR analysis: *urp1* forward: 5′-TCTGGCGGTGCTCTACATTC-3′, reverse: 5′-AGCAGGACAGGAAGCACAGT-3′; *urp2* forward: 5′- CCGGAGAACCAGATGCCTTT −3′, reverse: 5′-ATTTGGGCTGCTTGTTGCTG-3′. Relative gene expression levels were quantified using the comparative Ct method ($2^{-\Delta\Delta Ct}$ method) based on Ct values for target genes and zebrafish *β-actin*.

## Fluorescent beads tracing experiments

To evaluate hydrocephalus in zebrafish larvae, Rhodamine- or FITC-conjugated fluorescent dye (70 kDa) were injected directly into the brain ventricles at 2 or 3 dpf. Briefly, *ccdc57* mutants and control siblings were first anesthetized using 0.01% tricaine and then incubated with 20 mM 2,3-butanedione monoxime (BDM, Sigma) for 6 minutes to stop the heart beating. The larvae were then placed on the surface of a 1% agarose plate and further injected with fluorescent dye. Fluorescence images were collected using Leica M165FC fluorescent microscope. The size of each brain ventricle was measured and quantified using ImageJ software.

For fine particle movement analysis, we injected 20 nm or 100 nm fluorescent beads (F8888, Thermo Fisher Scientific) into the central canal of 30 hpf (20 nm) or 3 dpf (100 nm) zebrafish larvae. The injection methods were similar to previously reported [39,40]. Briefly, 30 hpf or 3 dpf larvae were anesthetized using 0.4 mg/ml tricaine and then mounted in 1.5% low melting point agarose in the lateral position. The 20-nm or 100-nm fluorescent beads were injected into the center of the diencephalic ventricle. At 1 hour after injection, time-lapse images were acquired at room temperature on an inverted Leica DMI8 spinning disk confocal microscope equipped with an Andor iXon Life 888 EMCCD using a 40X water immersion objective (N.A. = 1.1). These images of the fluorescent beads in the rostral part of central canal were acquired at a frame rate of 10 Hz using Fusion software. The data were further analyzed with ImageJ software.

To examine the migration of fluorescent beads in central canal, we microinjected 100 nm fluorescent beads into the central canal at the position above the end of the yolk extension. The migrated distance of fluorescent beads was captured by THUNDER Imager Model Organism and quantified using ImageJ software.

### 3D reconstruction of the choroid plexus

The adult brain was dissected and fixed in 4% PFA overnight at 4˚C. After immunofluorescent staining described above, the dChP was imaged using a 40X water objective on a Leica Sp8 confocal microscope. For 3D reconstruction of the CP, we first performed image binarization for acquired image stacks using ImageJ software (version 1.52p) and then performed surface rendering of the binary stack using Imaris software (version 9.6.0) to obtain 3D model.

### Cell polarity analysis

ImageJ software was used to quantify cell polarity. For basal body angle analysis, angles were measured between a line made between the centers of the cell and basal bodies and the horizontal axis, as illustrated in Fig 3E. The displacement distance ratio was calculated by dividing the distance between the center of the cell and the center of the basal body by the distance from the cell center to the cell membrane. Cell orientation and centers were calculated by MorphoLibJ plugins, and the raw data were analyzed by Microsoft Excel software to generate graphs.

### Human tissue collection

Paraspinal muscles were collected from 46 female AIS patients with main thoracic curve during corrective surgery. Bilateral deep paraspinal muscle biopsies of $1.5 \times 1.5 \times 1.5$ cm$^3$ were collected at the apical vertebral of the main curve for all the subjects. All patients or their guardians provided informed consent for the tissue collection.

### Statistical analysis

Statistical analysis was performed using ImageJ, Microsoft Excel, or GraphPad Prism 6 software. All data were presented as mean ± SD as indicated in the figure legends. A value of $p < 0.05$ was considered statistically significant. The numerical data used in all figures are included in S1 Data.

### Supporting information

**S1 Fig. Spinal curvature severity analysis in wild type and *ccdc57* mutant zebrafish.** (**A**) Alizarin red staining results of wild type and *ccdc57* mutant. (**B**-**E**) Statistical analysis and distribution pattern showing Cobb angles measured from dorsal-ventral and medio-lateral curvatures in female and male *ccdc57* mutant zebrafish. Scale bar: 1 cm in panel A. The data underlying the graphs shown in the figure can be found in S1 Data.
(TIF)

**S2 Fig. Phenotypes of *ccdc57* mutant and age matched sibling zebrafish larvae.** (**A**) External images showing zygotic and maternal zygotic (MZ) *ccdc57* mutant zebrafish at indicated developmental stages. (**B**-**E'**) Confocal images showing cilia in the ear cristae (EC) (**B**-**B'**), nasal pit (NP) (**C**-**C'**), pronephric duct (PD) (**D**-**D'**), and spinal canal (SC) (**E**-**E'**) in 5 dpf wild type and *ccdc57* mutants as indicated. Cilia were visualized with anti-glycylated tubulin in green, and nuclei were counterstained with DAPI in blue. (**F**) Statistical analysis showing relative cilia length in cristae, pronephric duct, and spinal canal. (**G**) Differential interference contrast

(DIC) images of notochords in wild type and *ccdc57* mutants. The purple dotted lines mark the margin of the neighboring notochord cells. (**H**) Fluorescent images showing calcein staining of 17 dpf wild type and *ccdc57^(tft168N)* mutant zebrafish as indicated. (**I**) Fluorescent images showing the osteoblasts marked by *Tg(Ola.Sp7:NLS-GFP)* in 17 dpf wild type and *ccdc57* mutant. Scale bars: 1 mm in panel A; 1 mm in panel H. The data underlying the graphs shown in the figure can be found in S1 Data.
(TIF)

**S3 Fig. No obvious hydrocephalus in *ccdc57* mutant larvae.** (**A**, **B**) Fluorescent images showing brain ventricles as indicated by injection of Rhodamine- or FITC-conjugated fluorescent beads (70 kDa) into zebrafish larvae at developmental stages as indicated. (**C**, **D**) Statistical analysis of brain ventricle size as indicated in panels (**A**) and (**B**). Scale bars: 50 μm in panels A and B. The data underlying the graphs shown in the figure can be found in S1 Data.
(TIF)

**S4 Fig. Distribution of multiciliated ependymal cells in adult wild type zebrafish.** Confocal images showing the relative position of the ependymal layer and dChP (white dotted box) in the telencephalon. Multiciliated ependymal cells were enriched in the center region (yellow dotted box) and monociliated cells were located to the periphery. Cilia were visualized with anti-glycylated tubulin antibody (red), and nuclei (green) were counterstained with DAPI. Scale bars: 7.5 μm, 250 μm, and 7.5 μm from the top to the bottom.
(TIF)

**S5 Fig. Mutation of *ccdc57* results in planar polarity defects in dChP.** (**A**) Three-dimensional reconstruction of dChP showing the distribution of cilia (anti-glycylated tubulin, red) in wild type and *ccdc57* mutant zebrafish. (**B**) Confocal images showing cilia are enriched in the center of the dChP folds as suggested from the 3D images. Cilia were visualized with anti-glycylated tubulin in green, and nuclei were counterstained with DAPI in blue. (**C**) Confocal images showing the distribution of the basal bodies in dChP of wild type and *ccdc57* mutant. (**D**) Angular distribution of the basal bodies in wild type and *ccdc57* mutants. (**E**) Statistical analysis showing the displacement distance of the basal bodies in wild type and *ccdc57* mutants. Scale bars: 50 μm in panel A; 25 μm in panel B; 5 μm in panel C. The data underlying the graphs shown in the figure can be found in S1 Data.
(TIF)

**S6 Fig. Cytoskeleton defects of the ependymal cells in the absence of Ccdc57.** (**A**) Confocal images showing the microtubule network visualized with anti-alpha tubulin antibody in green. (**B**) Diagram showing the disorganized cilia and microtubule skeleton in the ependymal cells of *ccdc57* mutant zebrafish. (**C**) Confocal images showing the distribution of mitochondria (Tom20) in wild type and *ccdc57* mutant. In panels (**A**) and (**C**), Phalloidin was used to mark the peripheral cortex of the cell. (**D**) Plot images showing the distribution pattern of mitochondria in wild type and *ccdc57* mutant. Scale bars: 7.5 μm in panel A; 5 μm in panel C. The data underlying the graphs shown in the figure can be found in S1 Data.
(TIF)

**S7 Fig. CSF flow defects in *ccdc57* mutants.** (**A**) Images showing the distribution of 100 nm fluorescent beads at 6 hour postinjection in wild type control or *ccdc57* mutant larvae at different developmental stages. Enlarged views of the magenta boxed regions are shown below each panel. Asterisks indicate injection sites; arrowheads indicate the migration end of fluorescent beads in the central canal. (**B**) Model illustrating the calculation methods of relative transport distance of the fluorescent beads in the spinal canal. (**C**) Statistical analysis showing relative

transport distances of injected fluorescent beads at 6 hour after injection. Scale bars: 10 μm in panel A. The data underlying the graphs shown in the figure can be found in S1 Data.
(TIF)

**S8 Fig. Rescue of body curvature defects through epinephrine treatments.** (**A**) Brightfield images of 3 dpf wild type and *ccdc57* mutants treated with DMSO or epinephrine. (**B**) Statistical analysis of body curvature angles in wild type and *ccdc57* mutants treated with DMSO or epinephrine. Scale bars: 1 mm in panel A. The data underlying the graphs shown in the figure can be found in S1 Data.
(TIF)

**S9 Fig. RF defects in cilia mutants.** (**A-C**) Confocal images showing the relative position of RF (WGA, red, white arrowhead) and cerebrospinal fluid-contacting neurons (CSF-cNs, green) marked with *Tg(urp1*:*GAL4; UAS*:*Kaede)* in 48 hpf wild type zebrafish larva. (**D-G**) Confocal images showing the RF (arrows) in different cilia mutants at 48 hpf. (**H-L'**) External image of different cilia mutants as indicated at 48 hpf and confocal images showing the RF (white arrows) of indicates mutants. (**M**) Pie chart displaying the statistical analysis of average curved angle in different mutants as indicated at 48 hpf. The RF were labeled with WGA in red, cilia were stained with anti-polyglutamylated tubulin in green, and nuclei were counterstained with DAPI in blue. Scale bars: 10 μm in panels A-C; 5 μm in panels D-G; 1 mm in panels H-L; 5 μm in panels H'-L'. The data underlying the graphs shown in the figure can be found in S1 Data.
(TIF)

**S10 Fig. RF assembly and Urotensin expression in *ccdc57* mutants.** (A) The bright field images of wild type and *ccdc57* mutants with different degree of scoliosis. (B) Confocal images showing the RF (white arrow indicated) in wild type and *ccdc57* mutants. The RF was labeled with WGA in red, and cilia were stained with anti-polyglutamylated tubulin in green. Nuclei were counterstained with DAPI in blue. (C) Schematic representation showing the position of the spinal cord that was dissected for in situ hybridization analysis. The hybridization results were shown on the right. Scale bars: 1 cm in panel A; 10 μm in panel B.
(TIF)

**S11 Fig. Multicilia were not essential for CSF flow and scoliosis progression.** (**A**) External phenotypes of the telencephalon in wild type and *gmnc* mutant. (**B**, **C**) Cross sections of the brain ventricles in wild type and *gmnc* mutant. (**D**) Representative images of 3-months-old wild type and *gmnc* mutants. (**E**) Kymographs of cilia movement in the spinal canal of 5 dpf wild type and *gmnc* mutant larvae. (**F**) Confocal images showing the RF (red, white arrowhead) in wild type and *gmnc* mutant larva. (**G**) Confocal images showing the distribution pattern of basal bodies in the ependymal cells of wild type and *gmnc* mutant. The basal bodies were labeled with anti-γ tubulin antibody in green, and tight junctions were stained with Claudin-5 antibody in red. (**H**) Confocal images and schematic graphs showing the distributed pattern of ependymal cells in wild type, *gmnc* and *ccdc57* mutants. Tight junctions were stained with Claudin-5 antibody in red. Scale bars: 100 μm in panel A; 1mm in panels B and C; 1 cm in panel D; 10 μm in panel F; 5 μm in panel G; 25 μm in panel H.
(TIF)

**S12 Fig. Slow progressive hydrocephalus formation due to CSF flow defects in *ccdc57* mutants.** (**A**) External images of wild type and *ccdc57* mutants at 17 dpf. (**B**) Cross-sections from the brain in wild type and *ccdc57* mutants. The white arrowhead indicates hydrocephalus in some *ccdc57* mutants. The number of dissected samples is shown in the bottom left. (**C**)

Bright field images showing the central canal labeled with purple pseudo-color in wild type and *ccdc57* mutants at different stages as indicated. (**D**) Bar graph with dots showing the width of central canal in wild type and *ccdc57* mutants at different developmental stages as indicated. Scale bar: 2 mm in panel A; 200 μm in panel B; 10 μm in panel C. The data underlying the graphs shown in the figure can be found in S1 Data.
(TIF)

**S1 Movie. 3D reconstruction of the diencephalic choroid plexus in wild type fish.** Posterior (P) is bottom.
(MP4)

**S2 Movie. 3D reconstruction of the diencephalic choroid plexus in *ccdc57* mutant.** Posterior (P) is bottom.
(MP4)

**S3 Movie. High-speed video microscopy showing multicilia beating in ependymal cells of wild type fish.** The video is replayed at 25 frames per second (fps).
(AVI)

**S4 Movie. High-speed video microscopy showing multicilia beating in ependymal cells of *ccdc57* mutant.** The video is replayed at 25 frames per second (fps).
(AVI)

**S5 Movie. Distribution of cilia labeled with anti-glycylated tubulin in the diencephalic choroid plexus of wild type fish.** Posterior (P) is bottom.
(MP4)

**S6 Movie. Distribution of cilia labeled by anti-glycylated tubulin in the diencephalic choroid plexus of *ccdc57* mutant. Posterior (P) is bottom.**
(MP4)

**S7 Movie. High-speed video microscopy showing cilium beating in the central canal of 5 dpf wild type zebrafish larva.** The video is replayed at 25 frames per second (fps). Anterior (A) is to the left; ventral (V) is bottom.
(AVI)

**S8 Movie. High-speed video microscopy showing cilium beating in the central canal of 5 dpf *ccdc57* mutant.** The video is replayed at 25 frames per second (fps). Anterior (A) is to the left; ventral (V) is bottom.
(AVI)

**S9 Movie. Representative movie of fluorescent beads movement (20 nm) observed in central canal of a 30 hpf WT embryo acquired with a spinning disk microscope at 10 Hz.** The video is replayed at 10 frames per second (fps). Anterior (A) is to the left; ventral (V) is bottom.
(AVI)

**S10 Movie. Representative movie of fluorescent beads movement (20 nm) observed in central canal of a 30 hpf *ccdc57* mutant embryo acquired with a spinning disk microscope at 10 Hz.** The video is replayed at 10 frames per second (fps). Anterior (A) is to the left; ventral (V) is bottom.
(AVI)

**S11 Movie. Representative movie of fluorescent beads movement (100 nm) observed in central canal of a 3 dpf WT embryo acquired with a spinning disk microscope at 10 Hz.** The video is replayed at 15 frames per second (fps). Anterior (A) is to the left; ventral (V) is bottom.
(AVI)

**S12 Movie. Representative movie of fluorescent beads movement (100 nm) observed in central canal of a 3 dpf *ccdc57* mutant embryo acquired with a spinning disk microscope at 10 Hz.** The video is replayed at 15 frames per second (fps). Anterior (A) is to the left; ventral (V) is bottom.
(AVI)

**S13 Movie. High-speed video microscopy showing cilium beating in the central canal of 5 dpf control zebrafish larva.** The video is replayed at 25 frames per second (fps). Anterior (A) is to the left; ventral (V) is bottom.
(AVI)

**S14 Movie. High-speed video microscopy showing cilium beating in the central canal of 5 dpf *gmnc* mutant.** The video is replayed at 25 frames per second (fps). Anterior (A) is to the left; ventral (V) is bottom.
(AVI)

**S1 Data. Excel spreadsheet containing, in separate sheets, the underlying numerical data and statistical analysis for Figs 2E, 3E, 3F, 3G, 4C, 4F, 5B, 5D, 5E, 5G, 5J, 6I, 6K, 7C, 7E, 7H, 8B, 8E, 9B, 9C, 9D, S1B, S1C, S1D, S1E, S2F, S3C, S3D, S5D, S5E, S6D, S7C, S8B, S9M and S12D.**
(XLSX)

**S1 Raw Images. The image shows the raw data for Fig 6E.**
(PNG)

## Acknowledgments

We thank the Goessling Lab for assistance in mapping the *ccdc57* gene mutation, Dr. Sudipto Roy for providing *gmnc* mutant, and Dr. Alexander F. Schier for providing GFP-Prickle construct. We thank Dr. Xin Liang, Dr. Xungang Tan, Dr. Xueliang Zhu, and Dr. Xiumin Yan for their kind help during the preparation of this manuscript. We are also grateful for the excellent support from the core facilities of Institute of Oceanology, Qingdao University and IEMB, OUC.

## Author Contributions

**Conceptualization:** Haibo Xie, Yunsi Kang, Chengtian Zhao.

**Data curation:** Haibo Xie, Yunsi Kang, Min Huang, Zhicheng Dai.

**Funding acquisition:** Leilei Xu, Pamela C. Yelick, Muqing Cao, Chengtian Zhao.

**Investigation:** Haibo Xie, Junjun Liu, Min Huang, Jiale Shi, Yuan Li, Yi Sun.

**Methodology:** Haibo Xie, Yunsi Kang, Min Huang, Lanqin Li, Pengfei Zheng, Muqing Cao, Chengtian Zhao.

**Project administration:** Chengtian Zhao.

**Resources:** Pamela C. Yelick, Chengtian Zhao.

**Software:** Shuo Wang.

**Supervision:** Chengtian Zhao.

**Validation:** Qize Han.

**Visualization:** Yunsi Kang, Shuo Wang.

**Writing – original draft:** Haibo Xie, Yunsi Kang, Zhicheng Dai, Muqing Cao, Chengtian Zhao.

**Writing – review & editing:** Jingjing Zhang, Zezhang Zhu, Leilei Xu, Pamela C. Yelick, Muqing Cao, Chengtian Zhao.

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
