## [Editor Report · Decision Letter 0]

7 Jul 2022

Dear Chengtian, 

Thank you for submitting your manuscript entitled "Spinal curvature due to defective planar polarity of the brain ependymal cells" for consideration as a Research Article by PLOS Biology.

Your manuscript has now been evaluated by the PLOS Biology editorial staff as well as by an academic editor with relevant expertise and I am writing to let you know that we would like to send your submission out for external peer review.

Once your full submission is complete, your paper will undergo a series of checks in preparation for peer review. After your manuscript has passed the checks it will be sent out for review. To provide the metadata for your submission, please Login to Editorial Manager (https://www.editorialmanager.com/pbiology) within two working days, i.e. by Jul 11 2022 11:59PM.

Kind regards,

Ines

--

Ines Alvarez-Garcia, PhD

Senior Editor

PLOS Biology

---

## [Decision Letter · Decision Letter 1]

22 Aug 2022

Dear Dr Zhao,

Thank you for your patience while your manuscript entitled "Spinal curvature due to defective planar polarity of the brain ependymal cells" was peer-reviewed at PLOS Biology. It has now been evaluated by the PLOS Biology editors, an Academic Editor with relevant expertise, and by three independent reviewers. 

The reviews are attached below. As you will see, the reviewers find the conclusions interesting and the manuscript worth pursuing for publication, but they also raise several points that need to be addressed. Reviewer 1 thinks you need to rewrite the Introduction to reflect the background literature better and also the Results and Discussion to acknowledge the Reissner fiber as the link between the flow and the sensory neurons CSF-cNs that express the peptides. This reviewer also asks you to analyse further if different mutations lead to different CSF flow and to provide quantification, along with establishing a correlation between the urp2 expression and the curvature of the spine. Reviewer 2 thinks the title should reflect better the findings and, like Reviewer 3, raises several points that need to be clarified.

In light of the reviews, we would like to invite you to revise the work to thoroughly address the reviewers' reports. Given the extent of revision needed, we cannot make a decision about publication until we have seen the revised manuscript and your response to the reviewers' comments. Your revised manuscript is likely to be sent for further evaluation by all or a subset of the reviewers.

**IMPORTANT - SUBMITTING YOUR REVISION**

3. Resubmission Checklist

a) *PLOS Data Policy*

b) *Published Peer Review*

d) *Blurb*

Please also provide a blurb which (if accepted) will be included in our weekly and monthly Electronic Table of Contents, sent out to readers of PLOS Biology, and may be used to promote your article in social media. The blurb should be about 30-40 words long and is subject to editorial changes. It should, without exaggeration, entice people to read your manuscript. It should not be redundant with the title and should not contain acronyms or abbreviations. For examples, view our author guidelines: https://journals.plos.org/plosbiology/s/revising-your-manuscript#loc-blurb

Sincerely,

Ines

--

Ines Alvarez-Garcia, PhD

Senior Editor

PLOS Biology

Reviewers' comments

Rev. 1:

In line with previous results indicating that mutants with defective motile cilia exhibit deformations of the body axis and spine (Grimes et al Science 2016) because their Reissner fiber does not form (Cantaut-Belarif et al CB 2018) or is not maintained properly (Troutwine et al CB 2020; Rose et al CB 2020), the authors had shown previously that mutants leading to defective CSF flow showed scoliosis due to defects in the URP signaling expressed by CSF-contact neurons and acting on a receptor in the dorsal musculature (Zhang et al Nature Genetics 2018).

Here the authors provide new interesting evidence for yet another cilia related gene, the gene ccdc57 shown to be involved in the planar polarity of cilia, to be linked via different mutations to early onset curvature of the body axis (3dpf) and 3D torsion of the spine (17 dpf) as well as hydroencephalus. The authors link the mutations in this gene to changes in urotensin signalling, and show the first evidence for difference on the concave and convex side of urotensin signalling (via expression of the receptor UTS2R in the musculature) of AIS patients.

Major comments

A) The authors do not currently mention in their introduction & most of their results the large body of work performed by their colleagues to establish critical role of the Reissner fiber with CSF-contacting neurons as the axial sensory system critically-linking cilia function and URP peptides : the authors only mention it at the bottom of page 12 !! this is not fair to the literature and our understanding of the mechanisms that all converge on the fiber to confer a straight body axis. The authors should therefore mandatorily fix this in their revised manuscript by properly introducing the large body of work of the last 4 years.

The authors absolutely need to revise the introduction, the results and discussion whose first paragraph should point to the Reissner fiber as the link between the flow and the sensory neurons CSF-cNs that express the peptides.

- the Reissner fiber needs to aggregate the CSF flow that originates from motile cilia with caudal polarity in the central canal (Cantaut-Belarif CB 2018) ; consequently the lack of the Reissner fiber in 6 mutants with defective cilia explains deformation of the body axis (Cantaut-Belarif CB 2018)

- CSF flow is sensed by CSF-cNs via their coupling to the fiber (Orts Dell Immagine et al CB 2020)

- Based on the coupling of the fiber and CSF-cNs, URP peptides are expressed in CSF-cNs to act on downstream targets outside of the nervous system (Cantaut-Belarif eLife 2020)

- Monoamines (epinephrine, norepinephrine) that can rescue defects in the body axis bind to the Reissner fiber to act (Cantaut-Belarif eLife 2020).

- Similar roles of the Reissner fiber established for straightening the body axis in the embryo appear conserved in the juveniles, as defects in maintenance of the Reissner fiber at the juvenile stage lead to 3D deformation (Troutwine et al CB 2020; Rose et al CB 2020; Lu et al Biol Open 2020).

B) Here the authors found another gene involved in ciliary function whose mutations can lead, depending of the severity, to either deformation of the body axis at 3dpf or of the spine at 17 dpf. We therefore need them to investigate further :

1) - whether the different mutations lead to different CSF flow (not only cilia beating) at 3 dpf or 17 dpf ***using the refined tracking of fluorescent beads*** :

this should be quantified as done previously in the embryo using injections of fluorescent beads (Thouvenin et al eLife 2020; Thouvenin et al Bio-protocol 2021, https://bio-protocol.org/exchange/protocoldetail?id=3932&type=0&searchid=1659333907627759&sort=0)

the authors should elaborate whether there are fine difference in flow, especially for the mutant alleles leading to defects at 3 or 17 dpf respectively.

2) - whether the severity of the morphogenetic defects are correlated with different level of defects in the formation or maintenance of the Reissner fiber - this is critical and not pursued enough,

3) - how to explain why the different mutations of ccdc57 lead to distinct effects on the CSF flow & on the Reissner fiber formation/maintenance.

4) - whether the effects on hydroencephalus was always correlated and possibly precedes or follows defects in the Reissner fiber formation/maintenance.

C) The authors establish a correlation between expression of urp2 and curvature of the spine.

1) They should quantify the segment number corresponding to the site of maximal deformation as well as the graded expression of the peptide.

2) In the same line, we would expect them to infer a novel mechanism at play to link local urp expression and curvature: this new evidence contrasts with the postulated action of URP acting broadly on the urts2r3 receptor in the dorsal musculature. How can the peptide have a restricted action if its target are widespread on the dorsal musculature ?

Minor comments

- stage of the fish should be given in mm as well as dpf in different facility / with different feeding protocol will not lead to the same stage

Rev. 2:

The focus of this manuscript by Xie et al is scoliosis developed in zebrafish ccdc57 mutants. In addition to morphological phenotypes, the authors provide evidence that the mutants show rotational, translation and tissue PCP defects in multi-ciliated ependymal cells.

Ccdc57 has been linked to centrosomes and cilia before. However, this is the first time ccdc57 is inked to scoliosis. Multiple additional interesting and novel results were presented, including hydrocephaly, disorganized cilia and planar polarity phenotypes at multiple levels in ccdc57 mutants. However, the manuscript is quite diffuse. The authors made multiple conclusions, yet the rationale of each experiment and the connection between different experiments is not always clear. For example, in zebrafish ciliary mutants, it is thought that changed urotensin expression leads to abnormal signaling through urotensin receptors specifically expressed in dorsal slow-twitch muscles. Here the authors show that in idiopathic scoliosis patients, the expression of UTS2R is higher in the convex side. Do the authors suggest that abnormal and somehow randomly sided expression of UTS2R is causal for scoliosis in these patients? This may be an interesting mechanism, yet how it relates to the rest of the manuscript is not clear. Another example is figure S9 and the discussion on gmnc. A previous study using a temperature sensitive mutant suggests that there is a time sensitive window for developing scoliosis in juvenile zebrafish. This time window coincides with a growth spur and the stage ccdc57 mutants develop significant axis phenotypes. In contrast, gmnc is mostly involved in the biogenesis of multi-cilia, which occurs in ependymal cells after this time window. In this context, the comparison between gmnc and ccdc57 mutants is not particularly revealing. As a result, a clear and unifying mechanism is lacking. A more focused approach and investigating a potential role of ccdc57 in the basal foot will strengthen this manuscript. Below is a list of specific points.

1. The title is very broad and does not convey the specifics of the manuscript well.

2. It was shown that flow is not defective in ccdc57 mutants at the embryonic stage. Is this still true at the stage when mutants start to develop scoliosis? This is significant because directional flow could entrain PCP.

3. Introduction: "We provide data showing that Ccdc57 is crucial for the establishment of planar polarity of ependymal cells, whose defects are probably the major reason of scoliosis formation in ciliary mutants". This statement seems overstretched.

4. Text regarding Fig 1D, "two alleles complemented each other" should be "two alleles failed to complement each other"

5. Text regarding different alleles of ccdc57 "Moreover, maternal Ccdc57 protein may contribute to early embryonic development in the tft168N mutants." Maternal protein will contribute to early development regardless genotype.

6. Figure 2. Please provide sample orientation for H and I, and explain H' and I'

7. Figure 3. Please provide the age of fish analyzed

8. Basal body localization and cilia orientation are not affected in ccdc57 mutants at larvae stages. Please discuss

9. Movies need better annotations with info regarding sample orientation, size scale and playback speed. Movie S7 shows almost pulsative movement of cilia, which is not obvious in S9. However, it is difficult to draw conclusions, especially for coordination between cilia, by visual inspection. Kymographs will be helpful.

10. Discussion: "Ccdc57 may orchestrate the polarity of the microtubule network downstream of PCP signals, and Ccdc57 mutations may cause abnormal organization of the microtubules and thereby disrupt proper cell polarity and the distribution of basal

bodies." Related to this, Figure S6 shows gross disorganization of microtubule, not necessarily the more subtle PCP of microtubules. It is also possible that defective basal foot disrupts the localization of basal bodies and subsequently the microtubule network.

Rev. 3:

This is a highly comprehensive look at the ccdc57 mutant zebrafish which exhibits phenotypically hydrocephalus and scoliosis. The down stream experimentation secondary to mutations within this animal model are well described including issues with the cilia, polarity and urotensin signaling. The article is well written and has significant figures within the main manuscript and supplemental material that support the overall interpretation that mutations within this gene cause ciliary defects resulting in differences in ciliary action, cellular polarity and potentially CSF flow. Phenotypically hydrocephalus and scoliosis develop in addition to a disorganization of the RF which has been shown to be of importance in spinal development.

I raise the following questions for the authors that could be addressed or clarified:

what is the role of ccdc57 within the mitochondria?

Relation between the disorganization of the RF and the Ccdc57 gene mutation causing ciliary issues. What element comes first in the formation of the spinal curvature?

There is significant detail as to when the scoliosis is observed within the mutants versus the wild type, but I could not find the timing in which the hydrocephalus appeared (see figure 2). does this happen simultaneously or at differing timepoints?

There are other models of scoliosis within the zebrafish - many with the associated hydrocephalus and RF disorganization. There is one model related to Kif7 that reportedly does not have either of these issues. This is also a ciliary protein with a dual function within the cilia and the hedgehog signaling pathway. Can the authors hypothesize this type of model that may exhibit scoliosis 'down stream' of other phenotypic observations?

---

## [Decision Letter · Decision Letter 2]

21 Dec 2022

Dear Dr Zhao,

Thank you for your patience while we considered your revised manuscript entitled "Ependymal polarity defects coupled with disorganized ciliary beating drive abnormal cerebrospinal fluid flow and spine curvature in zebrafish" for publication as a Research Article at PLOS Biology. This revised version of your manuscript has been evaluated by the PLOS Biology editors, the Academic Editor and one of the original reviewers.

Based on this review (attached below) and the Academic Editor's assessment of your revision, we are likely to accept this manuscript for publication, provided you satisfactorily address the data and other policy-related requests stated below.

We expect to receive your revised manuscript within two weeks. 

*Published Peer Review History*

*Press*

Sincerely,

Ines

--

Ines Alvarez-Garcia, PhD

Senior Editor

PLOS Biology

DATA POLICY:

Many thanks for submitting a file containing the raw data underlying the graphs shown in the figures. I have checked the file and I can't find the data for the following figures, thus please add it to the data file or let us know where the data is located:

Fig. 7H and Fig. S6D

In addition, please ensure that figure legends in your manuscript include information on where the underlying data can be found. For example, you can add to each of the corresponding figure legends: "The data underlying the graphs shown in the figure can be found in Data S1.

Reviewers' comments

Rev. 1: Claire Wyart - note that this reviewer has signed her review

The authors satisfactorily answered my requests for additional experiments and reformulation of the work in the large body of studies investigating the link between the Reissner fiber, ciliary functions and hydroencephalus.

Congratulations for this interesting study!

---

## [Editor Report · Decision Letter 3]

20 Jan 2023

Dear Dr Zhao,

Thank you for the submission of your revised Research Article entitled "Ependymal polarity defects coupled with disorganized ciliary beating drive abnormal cerebrospinal fluid flow and spine curvature in zebrafish" for publication in PLOS Biology. On behalf of my colleagues and the Academic Editor, Dagmar Wachten, I am delighted to say that we can in principle accept your manuscript for publication, provided you address any remaining formatting and reporting issues. These will be detailed in an email you should receive within 2-3 business days from our colleagues in the journal operations team; no action is required from you until then. Please note that we will not be able to formally accept your manuscript and schedule it for publication until you have completed any requested changes.

PRESS

Sincerely, 

Ines

--

Ines Alvarez-Garcia, PhD

Senior Editor

PLOS Biology
